# Immunoinformatics design of a novel multiepitope vaccine candidate against non-typhoidal salmonellosis caused by *Salmonella* Kentucky using outer membrane proteins A, C, and F

Elayoni E. Igomu[1]☯*, Paul H. Mamman[2]☯, Jibril Adamu[2]☯, Maryam Muhammad[3]‡, Abubarkar O. Woziri[2]‡, Manasa Y. Sugun[1]‡, John A. Benshak[4]‡, Kingsley C. Anyika[3]‡, Rhoda Sam-Gyang[1]‡, David O. Ehizibolo[5]☯

1 Bacterial Vaccine Production Department, National Veterinary Research Institute, Vom, Plateau State, Nigeria, 2 Department of Veterinary Microbiology, Ahmadu Bello University, Zaria, Kaduna State, Nigeria, 3 Bacterial Research Department, National Veterinary Research Institute, Vom, Plateau State, Nigeria, 4 Biotechnology Center, National Veterinary Research Institute, Vom, Plateau State, Nigeria, 5 Foot and Mouth Disease Department, National Veterinary Research Institute, Vom, Plateau State, Nigeria

☯ These authors contributed equally to this work.
‡ MM, AOW, MYS, JAB, KCA and RSG also contributed equally to this work.
* elayonigomu@gmail.com

## Abstract

The global public health risk posed by *Salmonella* Kentucky (*S.* Kentucky) is rising, particularly due to the dissemination of antimicrobial resistance genes in human and animal populations. This serovar, widespread in Africa, has emerged as a notable cause of non-typhoidal gastroenteritis in humans. In this study, we used a bioinformatics approach to develop a peptide-based vaccine targeting epitopes from the outer membrane proteins A, C, and F of *S.* Kentucky. Additionally, we employed flagellin protein (*fliC*) from *Salmonella* Typhimurium (*S.* Typhimurium) as an adjuvant to enhance the vaccine's effectiveness. Through this approach, we identified 14 CD8+ and 7 CD4+ T-cell epitopes, which are predicted to be restricted by various MHC class I and MHC class II alleles. The predicted epitopes are expected to achieve a population coverage of 94.91% when used in vaccine formulations. Furthermore, we identified seven highly immunogenic linear B-cell epitopes and three conformational B-cell epitopes. These T-cell and B-cell epitopes were then linked using appropriate linkers to create a multi-epitope vaccine (MEV). To boost the immunogenicity of the peptide construct, *fliC* from *S.* Typhimurium was included at the N-terminal. The resulting MEV construct demonstrated high structural quality and favorable physicochemical properties. Molecular docking studies with Toll-like receptors 1, 2, 4, and 5, followed by molecular dynamic simulations, suggested that the vaccine-receptor complexes are energetically feasible, stable, and robust. Immune simulation results showed that the MEV elicited significant responses, including IgG, IgM, CD8+ T-cells, CD4+ T-cells, and various cytokines (IFN-γ, TGF-β, IL-2, IL-10, and IL-12), along with a noticeable reduction in antigen levels. Despite these promising *in-silico* findings, further validation

**Data Availability Statement:** All relevant data are within the manuscript.

**Funding:** The author(s) received no specific funding for this work.

**Competing interests:** The authors have declared that no competing interests exist.

through preclinical and clinical trials is required to confirm the vaccine's efficacy and safety.

## 1.0 Introduction

Antibiotics play an indispensable role in modern medicine, serving as a critical treatment for a range of infectious diseases, and as a key component of supportive therapy in cancer care. They are essential after surgical procedures and are also crucial in livestock and food animal production [1]. However, the widespread and inappropriate use of antibiotics has significantly fueled the rise of antibiotic resistance in pathogenic bacteria, bringing us closer to the grim reality of a post-antibiotic era. It is predicted that by 2050, deaths from antimicrobial-resistant bacteria could surpass those from cancer [1].

Fluoroquinolone-resistant non-typhoidal *Salmonella*, along with third-generation cephalosporin-resistant and carbapenem-resistant *Enterobacter* species, are listed by the World Health Organization (WHO) as top-priority pathogens requiring urgent attention. These pathogens pose a significant public health threat due to their high levels of antimicrobial resistance, which complicate treatments and heighten the risk of widespread outbreaks [2]. Particularly concerning is the high-level antimicrobial resistance observed in *Salmonella enterica* subsp. *enterica* serovar Kentucky (*S.* Kentucky). The extensive zoonotic potential of *S.* Kentucky, along with its detection in various animal species and humans, underscores its status as a high-risk, globally significant multidrug-resistant (MDR) pathogen [3, 4].

The rise and continued spread of MDR *S.* Kentucky has been increasingly noted by health organizations and nations, with its association with high-level fluoroquinolone resistance gaining global epidemiological significance [5]. For severe non-typhoidal *Salmonella* infections in humans and animals, ciprofloxacin and third-generation cephalosporins are the preferred treatments. However, the resistance of *S.* Kentucky to these key antibiotics has significantly narrowed the available treatment options [3].

Vaccines have been instrumental for decades in preventing diseases and have had an unprecedented impact on human and animal health globally. Unlike antibiotics, vaccines are less likely to contribute to antimicrobial resistance (AMR), as vaccines and antibiotics work in fundamentally different ways [6, 7]. Vaccines are primarily used as a preventive measure, taking action before the invading bacterial pathogen can multiply. This preemptive approach reduces both the pathogen load and the extent of tissue invasion, substantially lowering the chances of AMR developing or AMR mutations occurring [8].

Over several years, *Salmonella* outer membrane proteins (OMPs) have been explored as potential vaccine candidates. The *Salmonella* OMPs, including outer membrane protein A (OmpA), outer membrane protein C (OmpC), and outer membrane protein F (OmpF), are widely distributed and conserved across various *Salmonella* serovars. This conservation suggests their potential use in diagnostics or as components of subunit or conjugate vaccines [9, 10]. Several studies have used crude preparation of *Salmonella* OMPs to evoke a strong immunological response, thereby conferring protective immunity against an array of *Salmonella spp.* [10]. Notably, OMPs possess surface-exposed epitopes that are easily recognized by T-cell and B-cell receptors [11, 12].

Recently, the WHO emphasized the urgent need for new vaccines targeting infectious bacterial pathogens in Sub-Saharan Africa, with a particular focus on non-typhoidal salmonellosis (NTS) [13, 14]. This region is especially vulnerable to the emergence of antimicrobial

resistance due to several factors, including unregulated antibiotic access, indiscriminate use in livestock and food animal production, and cross-border livestock trade. These practices have created significant selective pressure, leading to the rise of resistant bacterial clones [3, 15–17]. Furthermore, the lack of comprehensive disease surveillance and monitoring systems on the continent further complicates treatment protocols and hinders effective containment strategies, making it difficult to address the spread of these resistant pathogens [3, 15–17].

In response to the WHO's call, our study focuses on developing a multi-epitope vaccine (MEV) candidate to target NTS caused by *S.* Kentucky. Using computational immunological methods, we retrieved sequences of OmpA, OmpC, and OmpF from the National Center for Biotechnology Information (NCBI) databank to guide our vaccine design.

## 2.0 Materials and methods

See Fig 1.

### 2.1 Retrieval of OMPs of *Salmonella* Kentucky sequences

Amino acid sequences for OmpA, OmpC, and OmpF of *S.* Kentucky strains were retrieved in FASTA format from the NCBI database (https://www.ncbi.nlm.nih.gov/protein/). The accession numbers for these OMPs were noted. To refine the selection of OMPs specific to various *S.* Kentucky strains, the retrieved sequences were subjected to a BLAST analysis, with the query parameters set to 99% to 100% identity (BLAST: https://blast.ncbi.nlm.nih.gov). Additionally, the amino acid sequence in FASTA format of the flagellin protein (*fliC*) (GenBank accession number KAA0422324.1) from *Salmonella enterica subsp. enterica* serovar Typhimurium (*S.* Typhimurium) was selected as an adjuvant to enhance the efficacy of the vaccine peptide [18, 19].

### 2.2 Multiple sequence alignment of OMPs, consensus sequence generation, and diversity analysis

Generating a consensus sequence for each class of OMP was paramount to ensure protein residue conservation. The FASTA sequences for the various OMPs were aligned using Jalview version 2.11.2.7, a free cross-platform software for multiple sequence alignment editing, visualization, and analysis (https://www.jalview.org/) [20]. The sequences for OmpA, OmpC, and OmpF obtained from the NCBI database were manually entered into Jalview, which automatically aligns and generates predicted consensus sequences using BLOSUM62 parameters [21].

To analyze the diversity among OMPs, we examined the phylogenetic relationships among the different classes of OMPs using the Phylogeny.fr platform (http://phylogeny.lirmm.fr/phylo_cgi/index.cgi). Phylogeny.fr provides a streamlined pipeline that automatically configures and integrates several bioinformatics tools to generate a robust phylogenetic tree from a given set of FASTA sequences. In our study, we utilized the "One Click" mode of the server (http://phylogeny.lirmm.fr/phylo_cgi/simple_phylogeny.cgi). This mode uses MUSCLE 3.7 for multiple sequence alignment, Gblocks 0.91b for automatic alignment and curation, PhyML 3.0 for maximum likelihood-based phylogenetic inference, and TreeDyn 198.3 for tree visualization [21–23].

### 2.3 Epitope mapping

To identify the major histocompatibility complex (MHC) epitopes for vaccine design, we analyzed the consensus sequence of the different OMP classes using the Immune Epitope

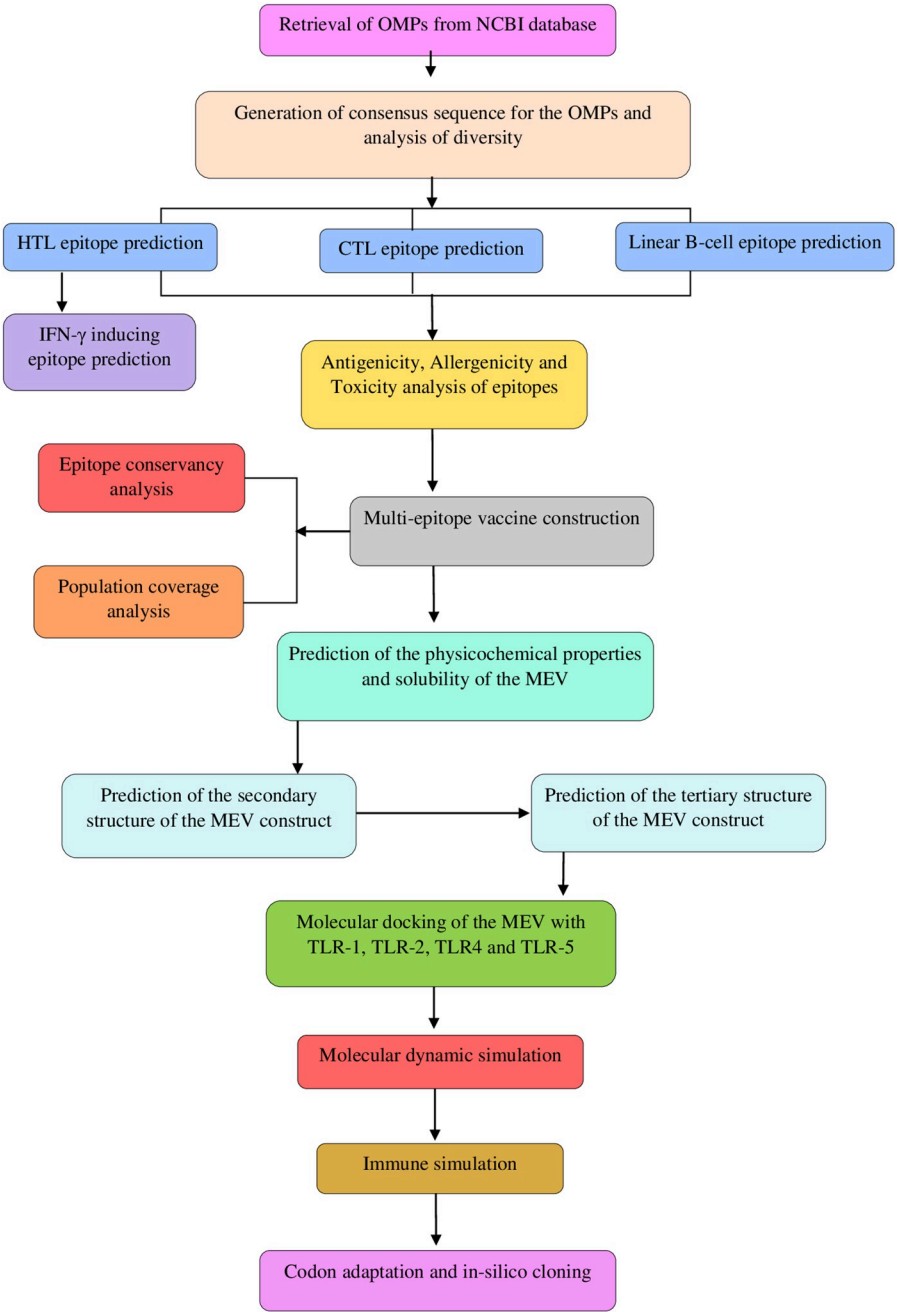

**Fig 1. Schematic representation of the study workflow for the construction and evaluation of the MEV.**

Database and Analysis Resource (IEDB) server (http://tools.iedb.org/main/tcell). This server predicts potential T-lymphocyte (T-cell) epitopes based on MHC binding data for both MHC class I (MHC-I) and MHC class II (MHC-II) [18, 24, 25].

**2.3.1 Cytotoxic T-lymphocyte epitopes prediction.** Cytotoxic T-lymphocytes (CTL) also called cytotoxic T-cells or CD8+ T-cells mostly express T-cell receptors (TCRs) capable of recognizing distinctive antigens. Thus, Identifying CTL epitopes is integral to designing a MEV. The prediction of CTL activating epitopes that bind to MHC-I was predicted on the IEDB

recommended 2023.09 (NetMHCpan4.1 EL) server (http://tools.iedb.org/mhci/). This server utilizes the weight matrix of the MHC-I binding peptides, the transporter associated with antigen processing (TAP), and the proteasomal C-terminus cleavage score for its prediction. The MHC source species was set as human and the human leucocyte antigen (HLA) allele reference set was predicted with a 9-mer epitope length. The threshold for the classification of epitopes as strong or weak binders was set at 0.5% and 0.2% respectively [25, 26].

**2.3.2 Helper T-lymphocyte epitopes prediction.** A critical component of the cellular and humoral immune system are helper T-lymphocytes (HTL) also called CD4+ T-cells. They are key regulators of a cascade of inflammatory processes necessary to subdue infection [27] and therefore form an integral part of MEV design. The prediction of HTL activating epitopes that bind to MHC-II was predicted on the IEDB recommended 2023.09 (NetMHCpan4.1 EL) server (http://tools.iedb.org/mhcii/). This server provides a ranked list of peptides based on calculations of IC50 values for the predicted epitopes, which are inversely related to their binding affinity to MHC-II molecules. IC50 values below 50 nM indicate high binding affinity, values between 50–500 nM suggest intermediate binding affinity and values above 500 nM indicate low binding affinity of the epitopes to MHC-II. The 7-allele HLA reference set was utilized to predict epitopes, with HLA-DR selected specifically for *Homo sapiens* on the server. Subsequently, a 15-mer length epitope with a percentile rank classification was obtained using the IEDB recommended 2023.05 method (NetMHCIIpan 4.1L). Only peptides with IC50 values below 50 nM were selected as epitopes in our vaccine design [24, 26].

## 2.4 Prediction of linear B-lymphocyte epitopes

The initiation of humoral immune responses requires antigen-reactive B-lymphocytes (B-cells) to encounter an antigen. B-cells, unlike T-cells, have B-cell receptors (BCRs), which bind to foreign antigens and trigger an antibody response. These BCRs are highly specific, with all BCRs on a single B-cell recognizing the same epitope. B-cell epitopes were predicted using the ABCpred server (https://webs.iiitd.edu.in/raghava/abcpred/ABC_submission.html), a neural network algorithm. A sequence length of 16 amino acids and a binding score threshold of > 0.51 were employed to predict linear B-cell epitopes for the OmpA, OmpC, and OmpF proteins using their FASTA consensus sequences. Epitopes with higher scores were selected for vaccine construction. Additionally, the Linear B-cell epitopes were further assessed for antigenicity using the antigen sequence properties tool on the IEDB website (http://tools.iedb.org/bcell/result/) [18, 19].

## 2.5 Prediction of antigenicity, allergenicity, and toxicity

The antigenicity of each predicted CTL, HTL, and linear B-cell epitope, as well as the constructed MEV sequence, was evaluated using the VaxiJen v2.0 server (http://www.ddg-pharmfac.net/vaxijen/VaxiJen/VaxiJen.html), with a bacterial threshold set at 0.4. This server employs auto and cross-covariance (ACC) transformation to generate its output [28–30].

To evaluate the potential allergenicity of the determined epitopes, the AllerTOP v2.0 server (http://www.ddg-pharmfac.net/AllerTOP) was utilized. AllerTOP also uses ACC transformation to convert protein sequences into uniform, equal-length vectors. It then compares the E descriptors of amino acids to obtain the closest k value. These k values are obtained based on a training set of 2,210 known allergens from various species and 2,210 non-allergens from the same species [31, 32].

Additionally, the predicted CTL, HTL, and linear B-cell epitopes were analyzed for potential toxicity using the ToxinPred server (https://webs.iiitd.edu.in/raghava/toxinpred/multi_submit.php), with default parameters. ToxinPred uses several algorithms, including

Quantitative Matrix (QM) methods, motif-based methods, and machine learning models such as Support Vector Machines (SVM), Random Forests (RF), and Decision Trees (DT), to predict peptide toxicity [25, 29].

## 2.6 Interferon-gamma-inducing epitope prediction

Interferon-gamma (IFN-γ) is critical for pathogen recognition and elimination, serving as a central effector of cell-mediated immunity [33]. This cytokine induction is a vital consideration for designing a MEV peptide against an infectious disease. The presence of IFN-γ inducing epitopes in the proposed MEV peptide was predicted using the IFNepitope server (http://crdd.osdd.net/raghava/ifnepitope/index.php). The server, trained with MHC-II epitopes, generates all possible overlapping peptides from an antigen based on IFN-γ inducing and non-inducing datasets, employing an SVM prediction system. The 15-mer MHC-II epitopes were submitted to the IFN-γ epitope server to assess their potential to induce an IFN-γ immune response. A hybrid motif and SVM model were selected from the available algorithm models, with a prediction score threshold set at >1.0 for epitope selection. Ultimately, high-response IFN-γ epitopes were selected for the MEV [34, 35].

## 2.7 Epitope conservancy analysis

Epitope conservancy is the degree to which a particular epitope sequence is preserved across different strains or species of a pathogen. The conservation of epitopes was predicted using the IEDB Conservancy Analysis tool (http://tools.iedb.org/conservancy/), a web-based tool designed to analyze epitope conservancy across various protein sequences [36]. Predicted CTL and HTL epitope sequences, along with their corresponding OMP class sequences, were entered into their respective text areas and the minimum conservancy sequence identity threshold was set at ≥ 100% [37].

## 2.8 Population coverage analysis of MHC-I and MHC-II epitopes

The IEDB's Population Coverage tool (http://tools.iedb.org/population/) was used to analyze how the affinity of CTL and HTL epitopes for HLA alleles varies by race, location, and country and how it impacts our multiepitope peptide vaccine design. The IEDB Population Coverage tool estimates the proportion or distribution of individuals expected to exhibit a response to the specified epitopes according to their identified HLA profiles. Furthermore, it evaluates the mean number of epitope matches/HLA allele pairings acknowledged by the overall populace, alongside the maximum and minimum counts of epitope matches acknowledged by 90% of the chosen population. Using the default parameters, 16 geographical areas were selected for the population coverage analysis [38, 39].

## 2.9 Construction of the MEV sequence

For the final assembly of the MEV, we selected epitopes based on their high antigenicity, non-allergenicity, and non-toxicity. Various combinations and permutations of epitope classes, ranging from the N-terminal to the C-terminal, were randomly generated during the construction process. The configuration with the highest antigenicity score was selected as the final model. B-cell epitopes (OmpA, OmpC, and OmpF) and HTL epitopes were connected using a GPGPG linker, while CTL epitopes were linked with an AAY linker. To further enhance the immunogenicity of the MEV, the adjuvant *fliC* (GenBank accession number KAA0422324.1) from *S.* Typhimurium was incorporated at the N-terminal of the epitopes via an EAAAK linker [40, 41].

## 2.10 Prediction of physicochemical characteristics and solubility

The MEV structure was analyzed on the Expasy ProtParam online server (https://web.expasy.org/protparam/) to predict various physicochemical properties of the multi-peptide arrangement, including the aliphatic index, molecular weight (MW), theoretical isoelectric point (pI), half-life, instability, and grand average of hydropathicity (GRAVY) [42, 43]. Additionally, the solubility value of the MEV was determined using the Protein-Sol server (http://protein-sol.manchester.ac.uk). A scaled solubility value exceeding 0.45 indicates a higher solubility profile compared to the average soluble *Escherichia coli* (*E. coli*) protein in the experimental solubility dataset. Conversely, proteins with lower-scaled solubility values are expected to be insoluble [44].

## 2.11 Secondary structure prediction

The secondary structure of a protein consists of locally folded formations within a polypeptide, driven by interactions among backbone atoms. These formations include α-helices, β-strands (which form β-sheets), and random coils. To determine the secondary structure of the MEV construct, predictions were made using the online tools PSIPRED and SOPMA. PSIPRED (http://bioinf.cs.ucl.ac.uk/psipred/) uses a neural network-based approach to assess the protein sequence and forecast the likelihood of each amino acid being part of an α-helix, β-strand, or coil [45]. SOPMA (https://npsa.lyon.inserm.fr/cgi-bin/npsa_automat.pl?page=/NPSA/npsa_sopma.html) similarly predicts secondary structures. Utilizing both PSIPRED and SOPMA allows for a more comprehensive and accurate prediction of a protein's secondary structure by capitalizing on the strengths of each tool [46].

## 2.12 Tertiary structure prediction

To accurately predict and model the three-dimensional (3D) configuration of the vaccine peptide and analyze the protein sequences' domains, the I-TASSER (Iterative Threading Assembly Refinement) software (https://zhanglab.ccmb.med.umich.edu/I-TASSER) was employed. This software is renowned for its accuracy in predicting 3D protein structures. It utilizes a sophisticated approach involving multiple threading alignments to identify template structures from the Protein Data Bank (PDB). Subsequently, it assembles full-length models by integrating structural fragments from these templates. Further refinement of the predicted models is achieved through iterative simulations to optimize their structure. The confidence score (C-score) is utilized to assess the accuracy of the 3D model, with higher values indicating superior quality, with the general range between -5 and 2 [19, 47].

## 2.13 Refinement and validation of tertiary structure

Validating the tertiary structure is essential in vaccine development to identify potential flaws in the predicted model [48]. The GalaxyRefine web server (http://galaxy.seoklab.org/cgi-bin/submit.cgi?type=REFINE) was used to enhance the quality of the 3D structure. This tool uses a sophisticated routine that involves reassembly and molecular dynamics modulation to optimize local structural regions, particularly improving side-chain conformations and main-chain positioning. The iterative refinement process of the server utilizes a force field that balances attractive and repulsive forces, resulting in a more stable and realistic structure. The server produces several refined models, ranked by quality, and with metrics such as RMSD, MolProbity scores, and other structural indicators provided for assessment [47, 49].

The ProSA-web server was employed to assess the overall quality score of the precise input structure for 3D validation. Additionally, ERRAT (http://services.mbi.ucla.edu/ERRAT/) was

utilized to analyze unbonded inter-atomic interactions and high-resolution crystallographic structures [50, 51]. To validate the refined tertiary structure of the vaccine model, the PRO-CHECK server (https://saves.mbi.ucla.edu) and VADAR version 1.8 (http://vadar.wishartlab.com/index.html?) were used [52]. PROCHECK assesses protein structure quality by evaluating dihedral angles, bond lengths, bond angles, and geometric properties, providing detailed stereo-chemical statistics and comprehensive reports. The results include the Ramachandran plot, which shows the proportion of residues within favored, permitted, and disallowed regions [53].

## 2.14 Prediction of discontinuous B-cell epitopes

The folding of proteins can bring distant residues into proximity, leading to the formation of discontinuous B-cell epitopes. Predicting these epitopes involves calculating the Protrusion Index (PI) and analyzing clustering based on the distance R, which measures the distance between the center of mass of residues. A higher R-value suggests a greater likelihood of encountering discontinuous epitopes. Given that over 90% of B-cell epitopes are discontinuous, the ElliPro server (http://tools.iedb.org/ellipro/) was used to predict these epitopes within the vaccine structure [25, 54]. We applied the default parameters, including a minimum residue score of 0.5 and a maximum distance of 6 Å. The server assesses the 3D structure of the vaccine, assigning an ellipsoid score to each residue based on the PI value. A PI score of 0.9 indicates that 90% of the protein's residues are within the ellipsoid, while 10% are outside. This PI value is determined by evaluating the center of mass of each residue relative to the largest feasible ellipsoid [25, 30].

## 2.15 Protein-protein docking

The vaccine model is expected to bind with immunological receptors to elicit an immune response. Toll-like receptors (TLRs) such as TLR-1, TLR-2, TLR-4, and TLR-5 are crucial for recognizing *Salmonella* structures and triggering production of inflammatory cytokines due to their high sensitivity to bacterial components, such as triacylated lipoproteins (TLR-1), lipoproteins (TLR-2), lipopolysaccharide (TLR-4), and flagellin (TLR-5) [55–57]. Thus, these receptors were selected for docking with the chimeric vaccine construct.

The potential of the chimeric vaccine construct docking with a TLR was evaluated by the ClusPro 2.0 server (https://cluspro.bu.edu/login.php). ClusPro 2.0 is a rigid-body protein-protein docking service that predicts interactions between two proteins. This software is entirely automated and employs 3 distinct procedures: the first is a unique fast Fourier transform (FFT) correlation, the second is clustering the best energy conformations, and the third is evaluating cluster stability using brief Monte Carlo simulations [47, 58, 59]. The PDB file of the refined vaccine construct was docked against TLR-1 (PDB Id: 6NIH), TLR-2 (PDB ID: 6NIG), TLR-4 (PDB ID: 3FXI), and TLR-5 (PDB ID: 3JOA) from the PDB database (https://www.rcsb.org/). The best-docked model was selected based on its lowest energy value among the models generated by the ClusPro 2.0 server.

## 2.16 Molecular dynamics simulation and protein-protein binding affinity analysis

To investigate the stability and dynamics of the docked complexes between MEV-TLR-1, MEV-TLR-2, MEV-TLR-4, and MEV-TLR-5, molecular dynamics simulations were conducted using the iMODS server (available at http://imods.chaconlab.org). This tool employs normal mode analysis (NMA) in internal coordinates to mimic the natural motions of biological macromolecules and generate plausible transition pathways between two similar structures,

even with large molecules [60]. Additionally, the Prodigy web server, accessible at https://wenmr.science.uu.nl/prodigy/, was utilized to predict the binding affinity and dissociation constant of the docked proteins. Default parameters were applied for all analyses [61–63].

## 2.17 Immune simulation

The vaccine amino acid sequence was input into the C-ImmSim server, accessible at https://kraken.iac.rm.cnr.it/C-IMMSIM/, to analyze the immune response profile in a computational model. The C-ImmSim server evaluates both humoral and cellular responses to the vaccine model [64] using a Position-specific scoring matrix (PSSM) and machine learning techniques. PSSM simulates various anatomical regions in mammals, including the Bone marrow (which stimulates hematopoietic stem cells and myeloid cell production), the thymus (where naive T-cells are selected to prevent autoimmunity), and tertiary lymphatic organs, to mimic real-life immune responses. Immune stimulation was conducted by injecting the designed peptide vaccine, with three shots administered at four-week intervals (days 0, 28, and 56), translating to time steps at 1, 84, and 168 for each injection. Each time step represents an 8-hour interval, with the initial infusion administered at time zero [64].

First dose, Day 0 (Time step 1): The first dose is administered to prime the immune system. This step initiates the primary immune response, where the body recognizes the antigen and begins producing antibodies and memory cells. This initial exposure is crucial for training the immune system to recognize the pathogen effectively. Second dose, Day 28 (Time step 84): The second dose, given four weeks after the first, enhances the immune response. This secondary exposure leads to more robust and quicker production of antibodies, as memory cells from the first dose are reactivated and proliferate. This step is essential for establishing a stronger and longer-lasting immunity. Third dose, Day 56 (Time step 168): The third dose, administered four weeks after the second, further strengthens the immune response. This dose helps solidify the memory of the pathogen in the immune system, ensuring long-term protection. The third dose significantly increases antibody titers and the durability of the immune response, providing a solid foundation for long-term immunity [65]. This prime-booster-booster strategy at 4-week intervals, was aimed to achieve a durable protective immune response. It was administrated with 336 simulation steps, with all other simulation parameters set to default values [19, 39, 65].

## 2.18 Codon adaptation

Ensuring the efficacy of the vaccine model during cloning and expression is vital for its evaluation in both *in-vitro* and *in-vivo* systems. *In-vitro* studies are particularly important in human vaccine development, as they provide essential preliminary data on safety and efficacy [66, 67]. Since the constructed MEV is intended for human use, optimizing codon usage is necessary to enhance protein expression in mammalian cell lines. This optimization addresses the differences in codon preference between mammals and *Salmonella enterica* strains, aiming for successful expression in the target host.

The Java Codon Adaptation Tool (JCAT) server (http://www.prodoric.de/JCat) was utilized to perform reverse translation of the vaccine construct's protein sequence into a DNA sequence. The process avoided rho-independent transcription termination, prokaryotic ribosome binding sites, and restriction enzyme cleavage sites [68]. In JCAT, a codon adaptation index (CAI) score of 1.0 is considered ideal, although scores above 0.8 are generally satisfactory. Additionally, the sequence's guanine (G) and cytosine (C) base content was optimized to fall within the favorable range of 30% to 70% [68].

## 2.19 Restriction enzyme mapping and *in-silico* cloning into plasmid

To analyze restriction enzyme sites within the JCAT codon-optimized sequence of the vaccine construct, we used the NEBcutter Version 3.0.17 software (available at https://nc3.neb.com/NEBcutter/). NEBcutter V3.0 is specifically designed to support users in planning restriction digests and molecular cloning projects. By submitting a sequence, or selecting from a library of common plasmids, users can generate visual maps of restriction enzyme sites and simulate digests with chosen enzymes. In our study, we introduced HindIII and BamHI restriction sites at the N-terminal and C-terminal regions of the DNA sequence respectively, as optimized by the JCAT server. Additionally, a hexa histidine tag (6xHis tag) was incorporated just before the BamHI site at the C-terminal to facilitate protein purification [47]. For the cloning process, we employed the SnapGene software to insert our DNA sequence into the pcDNA3.1_CT-GFP expression vector (detailed at https://www.snapgene.com/plasmids/mammalian_expression_vectors/pcDNA3.1_CT-GFP) [18, 50, 68, 69].

The pcDNA3.1_CT-GFP plasmid offers several advantages for vaccine design due to its unique features. This plasmid is equipped with a potent cytomegalovirus (CMV) promoter that ensures high levels of gene expression, essential for producing sufficient amounts of vaccine antigens. The plasmid also features a green fluorescent protein (GFP) tag, which facilitates monitoring of transfection efficiency by allowing easy visualization of the antigen expression through GFP fluorescence. Additionally, the pcDNA3.1 plasmid system is known for its high transfection efficiency in mammalian cells, increasing the likelihood that a substantial number of cells will express the vaccine antigen [18, 68–71].

## 3.0 Results

### 3.1 Retrieval of *S.* Kentucky OMPs sequences, generation of consensus sequence, and sequence diversity analysis

The amino acid sequences for *S.* Kentucky OmpA, OmpC, and OmpF, along with their accession numbers, were sourced from the NCBI protein database in FASTA format. In total, 65 OMPs from *S.* Kentucky were retrieved, comprising 18 OmpA, 32 OmpC, and 15 OmpF (Table 1). The FASTA format sequences of all 65 OMPs classes (A, C, and F) were aligned separately by Jalview 2.11.2.7. Additionally, the FASTA sequence of *fliC* (GenBank accession number KAA0422324.1) from *Salmonella* Typhimurium, used as an adjuvant, was also retrieved (Table 2).

The results from Phylogeny.fr, processed by Gblocks 0.91b, reveal the following conservation patterns within different classes of OMPs: OmpA exhibited 94% conservation, with 349 of the original 371 positions retained as conserved. OmpC showed 78% conservation, with 323 of the original 414 positions conserved, and OmpF had 88% conservation, with 319 of the original 363 positions conserved. In comparing conservation across all three OMP classes (OmpA, OmpC, and OmpF), only 27% of the original 459 positions were conserved, with 127 positions retained, indicating that over 72% of the positions are variable across these classes. The phylogenetic analysis conducted using PhyML 3.0 on the Phylogeny.fr server, illustrated in Fig 2, revealed a distinct evolutionary relationship among the OMPs. Notably, OmpC and OmpF exhibited a closer evolutionary relationship with each other than either did with OmpA.

### 3.2 Results for CTL and HTL epitope predictions and epitope conservancy analysis

At a threshold of 0.5% for strong binders, and after screening for antigenicity, allergenicity, and toxicity, we selected 4 CTL epitopes for OmpA, 5 CTL epitopes for OmpC, and 5 CTL

**Table 1. Accession numbers of OMPs A, C, and F selected for MEV design, retrieved from the NCBI protein database.**

| S/No. | OmpA | OmpC | OmpF |
|---|---|---|---|
| 1 | ELX43285.1 | EDX45415.1 | CAH2847802.1 |
| 2 | EDZ19256.1 | EDZ22335.1 | CAH2871873.1 |
| 3 | EDX48073.1 | CAH2866635.1 | ESG84125.1 |
| 4 | EJQ7725175.1 | CAH2866410.1 | ESG88044.1 |
| 5 | EDF7307973.1 | ERN84404.1 | ESC06125.1 |
| 6 | EJW1777139.1 | ERN77197.1 | ELX43265.1 |
| 7 | EBL3626682.1 | ERN70439.1 | ERN84743.1 |
| 8 | ECV5604453.1 | ERN69676.1 | ERN79626.1 |
| 9 | EBM7532700.1 | ERN69269.1 | ERN78365.1 |
| 10 | EFL6925197.1 | ESG79344.1 | ERN73560.1 |
| 11 | EBW8725894.1 | ESG77524.1 | ERN62106.1 |
| 12 | ECI8791934.1 | ESC11174.1 | EDZ19170.1 |
| 13 | EGC5146150.1 | ELX37395.1 | EDX44525.1 |
| 14 | ESG79962.1 | ERN88917.1 | EJC3163576.1 |
| 15 | ESG88025.1 | ERN87695.1 | EBL4572010.1 |
| 16 | ESC13401.1 | ERN87644.1 | |
| 17 | CAH2847702.1 | ERN82780.1 | |
| 18 | CAH2871911.1 | ERN77612.1 | |
| 19 | | ERN77081.1 | |
| 20 | | ERN76635.1 | |
| 21 | | ERN74999.1 | |
| 22 | | ERN72553.1 | |
| 23 | | ERN70652.1 | |
| 24 | | ERN69005.1 | |
| 25 | | ERN68685.1 | |
| 26 | | ERN67044.1 | |
| 27 | | ERN66706.1 | |
| 28 | | ERN66672.1 | |
| 29 | | ERN66415.1 | |
| 30 | | ERN64040.1 | |
| 31 | | ERN62720.1 | |
| 32 | | ERN62687.1 | |

epitopes for OmpF for inclusion in the MEV construction. The results of these screenings, including corresponding allele(s) and positions, are detailed in Tables 3 and 4. Using the IEDB server, with the 7-allele reference set and specified thresholds, we also identified 4 HTL epitopes for OmpA, 2 HTL epitopes for OmpC, and 2 HTL epitopes for OmpF that met the criteria for antigenicity, allergenicity, and toxicity. These HTL epitopes were likewise included in the MEV construction, with their screening results and corresponding allele(s) and positions also presented in Tables 3 and 4.

The IEDB server conservancy analysis, with a protein sequence identity threshold of 100%, predicted a minimum and maximum identity of 100% for all CTL and HTL epitopes across the OMP classes A, C, and F. Specifically, the epitope 'IEYAITPEI' showed 100% minimum and maximum identity conservancy as an OmpA epitope but had a minimum and maximum identity of 33.33% when considered as an OmpC CTL epitope (Table 4).

**Table 2. Consensus sequences of OMPs A, C, and F, along with *fliC*, generated using Jalview version 2.11.2.7 for MEV development.**

| Protein Class | FASTA sequences/number of amino acids | Percentage AA residue conservation |
|---|---|---|
| OmpA | >Consensus sequence comprising 371 AA residues | 94 |
| | MKKTAIAIAVALAGFATVAQAAPKDNTWYAGAKLGWSQYHDTGFINNDGPTHENQLGAGAFGGYQV NPYVGFEMGYDWLGRMPYKGDNINGAYKAQGVQLTAKLGYPITDDLDVYTRLGGMVWRADTKSNV PGPGGASNKDHDTGVSPVFAGGIEYAITPEIATRLEYQWTNNIGDANTIGTRPDNGLLSVGVSYRFG QQEAAPVVAPAPAPAPEVQTKHFTLKSDVLFNFNKSTLKPEGQQALDQLYSQLSNLDPKDGSVVVLG FTDRIGSDAYNQGLSEKRAQSVVDYLISKGIPSDKISARGMGESNPVTGNTCDNVKPRAALIDCLAPD RRVEIEVKGVKDVVTQPQARRVEIEIEVKGVKDVVTQPQA | |
| OmpC | >Consensus sequence comprising 403 AA residues | 78 |
| | MKVKVLALLVPALLVAGAANAAEIYNKDGNKLDLYGKVDGLHYFSDDKGSDGDQTYMRIGFKGETQIN DQLTGYGQWEYQIQGNQTEGSGDSSTRVAFAGLLFFDAGSFDDGGNYGVTYDVTSWTDVLPEFGG GTYGADNFMQQRGNGYATYRNTDFFGFVDGLDFALQYQGKNGSVSGENTNGRSGLNQNGDGYGGSLTY AIGEGFSVGGAITTSKRTADQNNTANDRLYGNGDRATVYTGGLKYDANNIYLAAQYSQTYNAARFGTSN GSTPSTSYGFANKAQNFEVVAQYQFDFGLRPSVAYLQSKGKDISNGGGASYGDQDIVKYVDVGATYY FNKNMSTYVDYKINLLDKNDFTRDAGINTDDIVALGLVYQFIVALGMVYQFIATDDIVGVGLVYQF | |
| OmpF | >Consensus sequence comprising 363 amino acid residues | 88 |
| | MMKRKILAAVIPALLAAATANAAEIYNKDGNKLDLYGKAVGRHVWTTTGDSKNADQTYAQIGFKGET QINTDLTGFGQWEYRTKADRAEGEQQNSNLVRLAFAGLKYAEVGSIDYGRNYGIVYDVESYTDM APYFSGETWGGAYTDNYMTSRAGGLLTYRNSDFFGLVDGLSFGIQYQGKNQDNHSINSQNGDGV GYTMAYEFDGFGVTAAYSNSKRTNDQQDRDGNGDRAESWAVGAKYDANNVYLAAVYAETRNM SIVENTVTDTVEMANKTQNLEVVAQYQFDFGLRPAISYVQSKGKQLNGADGSADLAKYIQAGATYY FNKNMNVWVDYRFNLLDENDYSSSYVGTDDQAAVGITYQF | |
| *fliC* | 49 AA residues | |
| | MAQVINTNSLSLLTQNNLNKSQSALGTAIERLSSGLRINSAKDDAAGQA | |

Key: **AA:** Amino acids.

### 3.3 Linear B-lymphocyte epitopes prediction

The ABCpred server identified 35 B-cell epitopes for the consensus sequence of OmpA, and 39 and 41 epitopes for the consensus sequences of OmpC and OmpF, respectively, using a cutoff binding score of 0.51. From these, B-cell epitopes with a binding score of $\geq 0.85$ and a minimum peptide rank score of 7 were selected for further screening. These selected epitopes underwent additional antigenicity, allergenicity, and toxicity assessments. Ultimately, 2 B-cell epitopes for OmpA, 2 for OmpC, and 3 for OmpF were chosen for inclusion in the MEV construction (Table 5).

### 3.4 IFN-γ-inducing epitope prediction

Eight MHC-II epitopes were submitted to the IFNepitope server for IFN-γ prediction. Of these, two HTL epitopes were identified as positive inducers of IFN-γ production. The SVM method predicted the epitope 'LAPDRRVEI' as a positive inducer, while the MERCI method identified the epitope 'IEYAITPEI' as a positive inducer (Table 6).

### 3.5 Population coverage

Analysis of the MHC-I and MHC-II alleles confirmed that the intended vaccine architecture would provide adequate coverage for 94.91% of the global population. Continental and regional analyses revealed that Europe, North America, and North Africa had the highest coverage at 97.59%, 96.92% and 92.89% respectively (Fig 3).

In Africa, coverage percentages were as follows: West Africa 89.77%, South Africa 83.02%, East Africa 81.86%, and Central Africa 78.17%, with a regional average of 85.14% (Table 7). Outside the African continent, East Asia, North-East Asia and South America were 90.82%,

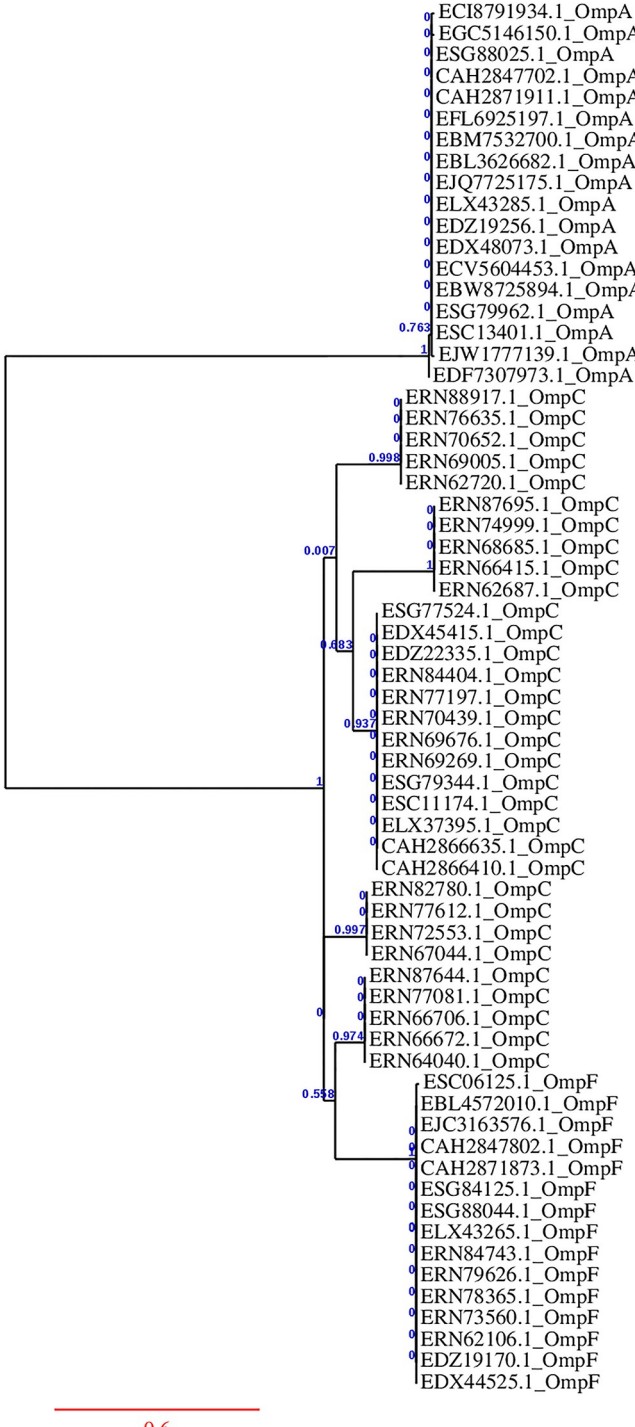

**Fig 2. Phylogenetic relationships among OMP classes were derived using maximum-likelihood method on the Phylogeny.fr server.**

**Table 3. Predicted CTL and HTL epitopes of *S*. Kentucky OMPs A, C, and F, along with their immunogenic properties.**

| Protein class | Epitopes | Peptide Score | Antigenicity (Score) | Allergenicity | Toxicity |
|---|---|---|---|---|---|
| OmpA | LSVGVSYRF | 0.93 | Antigenic (1.5084) | Non-Allergen | Non-Toxin |
| | EVQTKHFTL | 0.91 | Antigenic (1.5694) | Non-Allergen | Non-Toxin |
| | IEYAITPEI | 0.81 | Antigenic (1.2128) | Non-Allergen | Non-Toxin |
| | VQLTAKLGY | 0.84 | Antigenic (0.7491) | Non-Allergen | Non-Toxin |
| | FNFNKSTLK | 0.74 | Antigenic (0.6346) | Non-Allergen | Non-Toxin |
| | YKGDNINGA | 0.59 | Antigenic (1.2899) | Non-Allergen | Non-Toxin |
| | LAPDRRVEI | 0.89 | Antigenic (0.5152) | Non-Allergen | Non-Toxin |
| OmpC | VALGLVYQF | 0.91 | Antigenic (0.8219) | Non-Allergen | Non-Toxin |
| | GLRPSVAYL | 0.87 | Antigenic (0.9049) | Non-Allergen | Non-Toxin |
| | LPEFGGGTY | 0.89 | Antigenic (0.8301) | Non-Allergen | Non-Toxin |
| | ISNGGGASY | 0.82 | Antigenic (1.5949) | Non-Allergen | Non-Toxin |
| | RGNGYATYR | 0.80 | Antigenic (1.0838) | Non-Allergen | Non-Toxin |
| | FNKNMSTYV | 0.82 | Antigenic (1.1373) | Non-Allergen | Non-Toxin |
| | MSTYVDYKI | 0.81 | Antigenic (1.0144) | Non-Allergen | Non-Toxin |
| OmpF | SQNGDGVGY | 0.93 | Antigenic (2.5198) | Non-Allergen | Non-Toxin |
| | DENDYSSSY | 0.92 | Antigenic (0.6581) | Non-Allergen | Non-Toxin |
| | AAVGITYQF | 0.84 | Antigenic (1.2698) | Non-Allergen | Non-Toxin |
| | IQAGATYYF | 0.87 | Antigenic (0.6830) | Non-Allergen | Non-Toxin |
| | TAAYSNSKR | 0.91 | Antigenic (0.4845) | Non-Allergen | Non-Toxin |
| | YYFNKNMNV | 0.93 | Antigenic (0.8472) | Non-Allergen | Non-Toxin |
| | YAETRNMSI | 0.91 | Antigenic (0.9418) | Non-Allergen | Non-Toxin |

79.71% and 83.0% respectively (Fig 3). Additionally, the putative HLA class I restricted epitopes exhibited a higher individual percentage coverage (90.02%) when queried against the HLA class II restricted epitopes (49.02%) across the world (Fig 3).

## 3.6 Construction of MEV peptide sequence and prediction of IFN-γ induction potential

The MEV construct consists of 462 amino acids residues derived from 28 selected antigenic T-cells and B-cell epitopes, including 14 CTLS, 7 HTLs, and 7 B-cell epitopes, covalently linked to an immuno-adjuvant *fliC* (GenBank accession number KAA0422324.1, with 49 amino acid residues). The *fliC* was fused to the CTL epitopes using an EAAAK linker at the N-terminal end of the vaccine construct. The CTL epitopes from OmpA, OmpC, and OmpF were combined with an AAY linker, and this fusion was linked to the first HTL OmpA epitope, using an AAY linker. The HTL and B-cell epitopes were fused with a GPGPG linker, with the OmpF B-cell epitope positioned at the C-terminal end of the vaccine construct (Fig 4).

Predictive analyses of the MEV construct for antigenicity, allergenicity, and toxicity confirm that the vaccine is antigenic (with a score of 1.0458), non-allergenic, and non-toxic. The IFNepitope SVM hybrid model results confirmed its ability to induce IFN-γ, with a score of 1.5695871.

## 3.7 Prediction of physicochemical characteristics and solubility of the MEV

The physicochemical parameters and solubility properties of the chimeric vaccine construct using Expasy ProtParam and the Protein-Sol servers, expressed the following: The molecular weight of the vaccine construct was 47343.98 Da and the bio-computed theoretical isoelectric

**Table 4. The predicted MHC-restricted allele(s) of the CTL and HTL epitopes and degree of conservancy for the OMPs A, C, and F.**

| Protein Class | MHC-I Epitopes | MHC-II Epitopes | MHC Restricted Allele(s) | Start | End | Percent of protein sequence matches at identity < = 100% | Minimum identity | Maximum identity |
|---|---|---|---|---|---|---|---|---|
| **OmpA** | LSVGVSYRF | | HLA-B*58:01, | 117 | 125 | 100.00% (1/1) | 100.00% | 100.00% |
| | | | HLA-B*57:01 | | | | | |
| | EVQTKHFTL | | HLA-B*08:01 | 143 | 151 | 100.00% (1/1) | 100.00% | 100.00% |
| | IEYAITPEI | | HLA-B*40:01 | 82 | 90 | 100.00% (1/1) | 100.00% | 100.00% |
| | VQLTAKLGY | | HLA-B*15:01, | 26 | 34 | 100.00% (1/1) | 100.00% | 100.00% |
| | | | HLA-A*30:02 | | | | | |
| | | FNFNKSTLK | HLA-DRB3*02:02, HLA-DRB5*01:01 | 152 | 168 | 100.00% (1/1) | 100.00% | 100.00% |
| | | YKGDNINGA | HLA-DRB3*01:01, | 9 | 23 | 100.00% (1/1) | 100.00% | 100.00% |
| | | | HLA-DRB3*02:02 | | | | | |
| | | LAPDRRVEI | HLA-DRB1*03:01 | 254 | 270 | 100.00% (1/1) | 100.00% | 100.00% |
| | | IEYAITPEI | HLA-DRB1*07:01 | 79 | 93 | 100.00% (1/1) | 100.00% | 100.00% |
| **OmpC** | VALGLVYQF | | HLA-B*57:01, | 298 | 306 | 100.00% (1/1) | 100.00% | 100.00% |
| | | | HLA-B*58:01, | | | | | |
| | | | HLA-B*53:01, | | | | | |
| | | | HLA-B*35:01 | | | | | |
| | IEYAITPEI | | HLA-DRB1*07:01 | 79 | 93 | 0.00% (0/1) | 33.33% | 33.33% |
| | GLRPSVAYL | | HLA-A*02:03, | 226 | 234 | 100.00% (1/1) | 100.00% | 100.00% |
| | | | HLA-A*02:01 | | | | | |
| | LPEFGGGTY | | HLA-B*35:01 | 56 | 64 | 100.00% (1/1) | 100.00% | 100.00% |
| | ISNGGGASY | | HLA-B*15:01, | 241 | 249 | 100.00% (1/1) | 100.00% | 100.00% |
| | | | HLA-A*30:02, | | | | | |
| | | | HLA-A*01:01, | | | | | |
| | | | HLA-B*35:01 | | | | | |
| | RGNGYATYR | | HLA-A*31:01 | 73 | 81 | 100.00% (1/1) | 100.00% | 100.00% |
| | | FNKNMSTYV | HLA-DRB3*02:02 | 263 | 278 | 100.00% (1/1) | 100.00% | 100.00% |
| | | MSTYVDYKI | HLA-DRB1*15:01 | 265 | 281 | 100.00% (1/1) | 100.00% | 100.00% |
| **OmpF** | SQNGDGVGY | | HLA-B*15:01, | 117 | 125 | 100.00% (1/1) | 100.00% | 100.00% |
| | | | HLA-A*30:02 | | | | | |
| | DENDYSSSY | | HLA-B*44:03, | 268 | 276 | 100.00% (1/1) | 100.00% | 100.00% |
| | | | HLA-B*44:02 | | | | | |
| | AAVGITYQF | | HLA-B*58:01, | 283 | 291 | 100.00% (1/1) | 100.00% | 100.00% |
| | | | HLA-B*57:01, | | | | | |
| | | | HLA-B*35:01, | | | | | |
| | | | HLA-A*32:01 | | | | | |
| | IQAGATYYF | | HLA-A*23:01, | 244 | 252 | 100.00% (1/1) | 100.00% | 100.00% |
| | | | HLA-A*24:02, | | | | | |
| | | | HLA-B*15:01 | | | | | |
| | TAAYSNSKR | | HLA-A*68:01 | 137 | 145 | 100.00% (1/1) | 100.00% | 100.00% |
| | | YYFNKNMNV | HLA-DRB3*02:02 | 244 | 262 | 100.00% (1/1) | 100.00% | 100.00% |
| | | YAETRNMSI | HLA-DRB1*07:01 | 174 | 190 | 100.00% (1/1) | 100.00% | 100.00% |

point (pI) was 9.150. The estimated half-life in mammalian cell is 30 hours; in yeast, it is >20 hours; and in *E. coli* it is >10 hours. The solubility of the vaccine was found to be 0.452 on the protein sol server (value > 0.45 is predicted to have a higher solubility than the average soluble *E. coli* protein). The instability index of the vaccine was 25.66, signifying that the vaccine is

**Table 5. Predicted linear B-cells epitopes of *S*. Kentucky OMPs A, C, and F along with the adjuvant *fliC*, and their immunogenic properties.**

| Protein class | B-cell Epitopes (16mers) | Average K and T Score | Peptide score | Antigenicity (Score) | Allergenicity | Toxicity |
|---|---|---|---|---|---|---|
| OmpA | DYLISKGIPSDKISAR | 1.020 | 0.89 | Antigenic (0.8497) | Non-Allergen | Non-Toxin |
|  | GPGGASNKDHDTGVSP | 0.953 | 0.88 | Antigenic (1.7585) | Non-Allergen | Non-Toxin |
| OmpC | TSNGSTPSTSYGFANK | 0.980 | 0.90 | Antigenic (0.8885) | Non-Allergen | Non-Toxin |
|  | AGAANAAEIYNKDGNK | 0.971 | 0.89 | Antigenic (1.2510) | Non-Allergen | Non-Toxin |
| OmpF | VGRHVWTTTGDSKNAD | 0.956 | 0.92 | Antigenic (1.4812) | Non-Allergen | Non-Toxin |
|  | GFGVTAAYSNSKRTND | 0.994 | 0.90 | Antigenic (1.2497) | Non-Allergen | Non-Toxin |
|  | DGLSFGIQYQGKNQDN | 1.001 | 0.89 | Antigenic (1.1403) | Non-Allergen | Non-Toxin |
| *fliC* | GenBank accession number KAA0422324.1 |  |  | Antigenic (0.7058) | Non-Allergen | Non-Toxin |

Key: **K and T**: Kolaskar and Tongaonkar antigenicity scale.

**Table 6. Predicted OMPs A, C and F MHC-II T-cell epitopes capable of inducing IFN-γ.**

| Protein class | Epitope 15 length Peptide | Epitope Sequence | Method | Result | Score |
|---|---|---|---|---|---|
| OmpA | DVLFNFNKSTLKPEG | FNFNKSTLK | SVM | NEGATIVE | -0.3144975 |
|  | RMPYKGDNINGAYKA | YKGDNINGA | SVM | NEGATIVE | -0.073986311 |
|  | IDCLAPDRRVEIEVK | LAPDRRVEI | SVM | POSITIVE | -0.053153702 |
|  | AGGIEYAITPEIATR | IEYAITPEI | MERCI | POSITIVE | 1 |
| OmpC | FNKNMSTYVDYKINL | FNKNMSTYV | SVM | NEGATIVE | -0.96274966 |
|  | FNKNMSTYVDYKINL | MSTYVDYKI | MERCI | NEGATIVE | 1 |
| OmpF | GATYYFNKNMNVWVD | YYFNKNMNV | SVM | NEGATIVE | -0.62234638 |
|  | AAVYAETRNMSIVEN | YAETRNMSI | SVM | NEGATIVE | -0.55036682 |

stable in a solvent environment ($>$ 40 signifies instability). The computed aliphatic index of the protein is 57.23. The aliphatic index quantifies the relative volume occupied by the aliphatic side chains of specific amino acids: alanine, valine, leucine, and isoleucine. A higher aliphatic index is associated with increased thermo-stability in globular proteins. The GRAVY score for the vaccine proteins was calculated to be -0.429, indicating they have good interactions with water molecules. The GRAVY score is a common parameter used in codon usage bias (CUB) analysis to assess a protein's hydrophobic and hydrophilic properties. The GRAVY score ranges from -2 to 2, where positive values indicate hydrophobic proteins and negative values indicate hydrophilic proteins.

## 3.8 Prediction of secondary and tertiary structure of the MEV

The secondary structure analysis of the designed MEV, performed using the PSIPRED and SOPMA servers, indicated that 77 amino acids (16.67%) were alpha-helices, 41 amino acids (8.87%) were extended beta-strands, 21 amino acids (4.55%) were beta-turns, and 323 amino acids (69.91%) were random coils (Fig 5). The 3D structure was modeled by the I-TASSER server, which predicted five tertiary structures based on threading templates. The best model, with a C-score of -0.69 was selected for further refinement and validation. The consistency of this 3D model was improved using the GalaxyRefine web server. Five refined models were generated from the initial MEV model (the -0.69 C-score structure from I-TASSER). Model 2 was the most significant based on computational structural qualities, with a GDT-HA score of

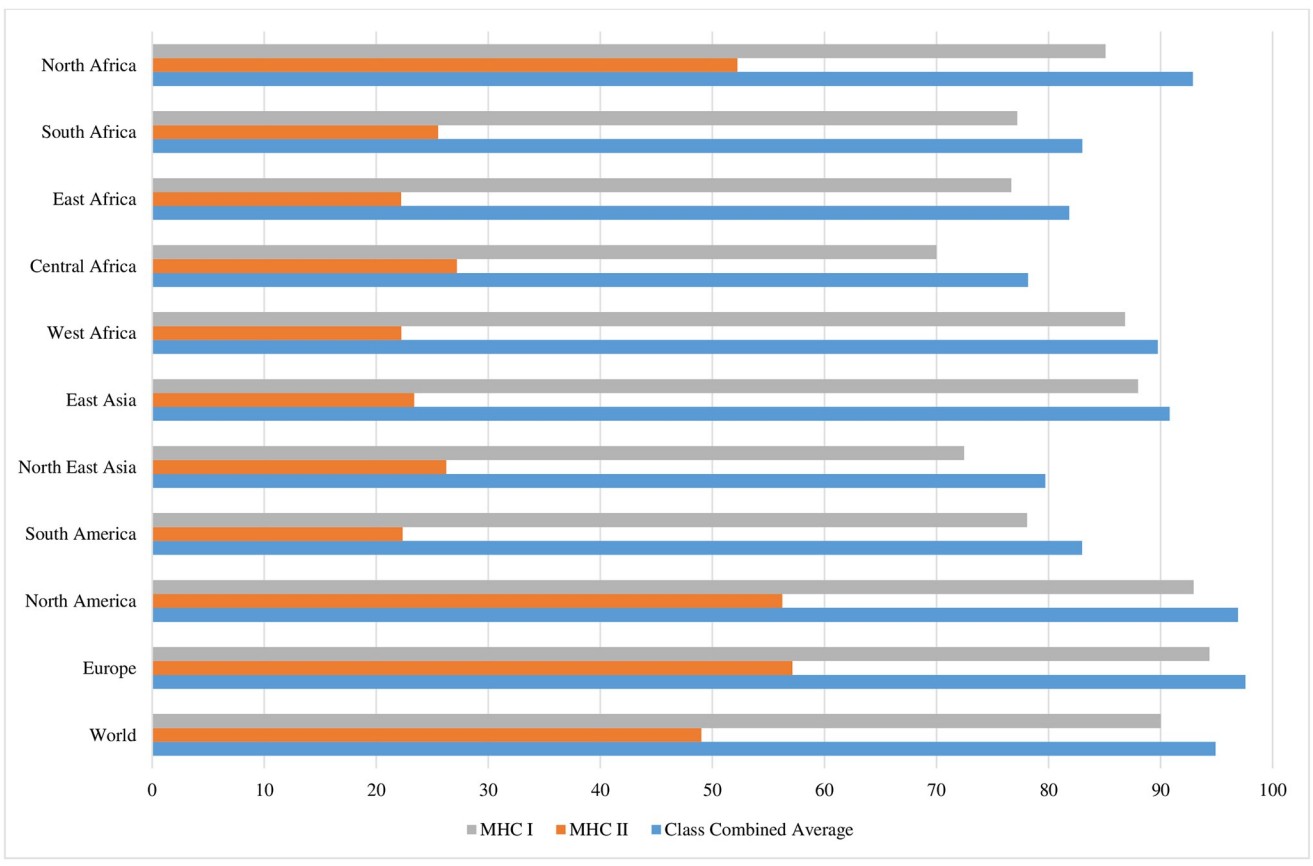

**Fig 3. Continental and regional coverage of the MHC-I and MHC-II T-Cell epitopes.**

0.9221, RMSD of 0.493, MolProbity score of 2.352, Clash score of 13.7, zero poor rotamers (0.0), and a Ramachandran favored value of 82.0 (Fig 6).

Validation of the refined tertiary structure of the MEV on the PROCHECK and VADAR version 1.8 servers produced a Ramachandran plot with 264 residues (75.6%) in the most

**Table 7. Percentage population coverage of MHC-I and MHC-II epitopes across various regions in Africa.**

| Population/Area | Class I | | | Class II | | | Class combined | | |
|---|---|---|---|---|---|---|---|---|---|
| | Coverage[a] | Average hit[b] | pc90[c] | Coverage[a] | Average hit[b] | pc90[c] | Coverage[a] | Average hit[b] | pc90[c] |
| Central Africa | 70.01% | 1.74 | 0.33 | 27.2% | 0.42 | 0.14 | 78.17% | 2.16 | 0.46 |
| East Africa | 76.69% | 1.88 | 0.43 | 22.21% | 0.32 | 0.13 | 81.86% | 2.2 | 0.55 |
| North Africa | 85.1% | 2.09 | 0.67 | 52.24% | 0.84 | 0.21 | 92.89% | 2.93 | 1.14 |
| South Africa | 77.2% | 1.86 | 0.44 | 25.52% | 0.26 | 0.13 | 83.02% | 2.12 | 0.59 |
| West Africa | 86.84% | 2.61 | 0.76 | 22.25% | 0.32 | 0.13 | 89.77% | 2.93 | 0.98 |
| Average | 79.17 | 2.04 | 0.53 | 29.88 | 0.43 | 0.15 | 85.14 | 2.47 | 0.74 |
| Standard deviation | 6.13 | 0.31 | 0.16 | 11.34 | 0.21 | 0.03 | 5.39 | 0.38 | 0.27 |

Key

[a] projected population coverage

[b] average number of epitope hits / HLA combinations recognized by the population

[c] minimum number of epitope hits / HLA combinations recognized by 90% of the population

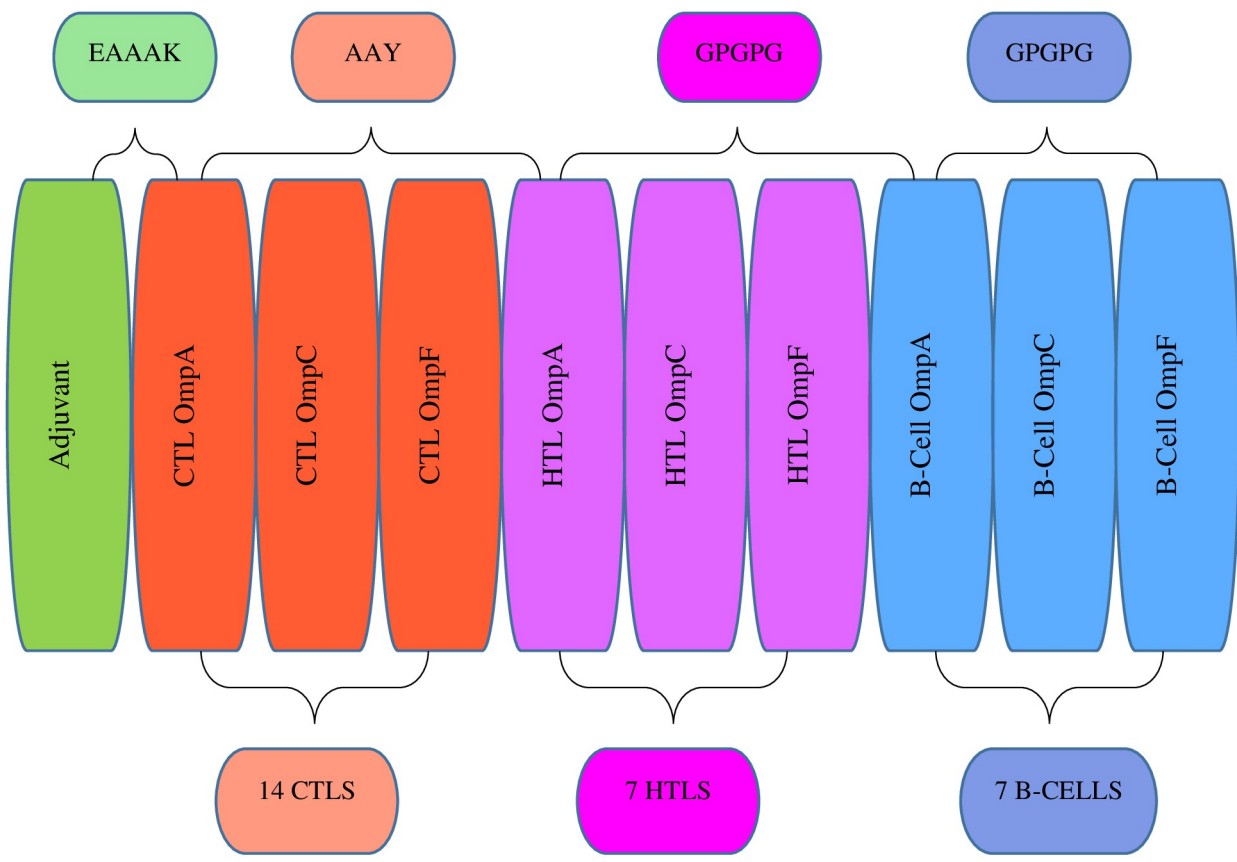

**Fig 4. The schematic structural arrangement of the MEV construct.**

favored region, 57 residues (16.3%) in additional allowed regions, 19 residues (5.4%) in generously allowed regions, and 9 number of residues (2.6%) in the disallowed regions, making a total of 349 number of non-glycine and non-proline residues. Additionally, there were 77 glycine residues, 34 proline residues, and 2 end-residues (excluding Glycine and Proline) (Fig 7).

### 3.9 Predicted discontinuous B-cell epitopes

The prediction of discontinuous epitopes requires the 3D structural information of the protein or polypeptide. Thus, ElliPro server was used to predict three discontinuous B-cell epitopes for the MEV, indicating its residues, number of residues, and scores (Table 8). The 3D representation of putative B-cell epitopes is shown in Fig 8, with the yellow surface highlighting them.

### 3.10 Protein-protein docking and protein-protein binding affinity simulation

Results from the ClusPro web server provided a protein-protein docking complex with the best model selected based on their lowest energy level score (Table 9). The 3D tertiary structural models of the selected TLRs-MEV docked complexes are in Fig 9. ChimeraX software was used for visualization.

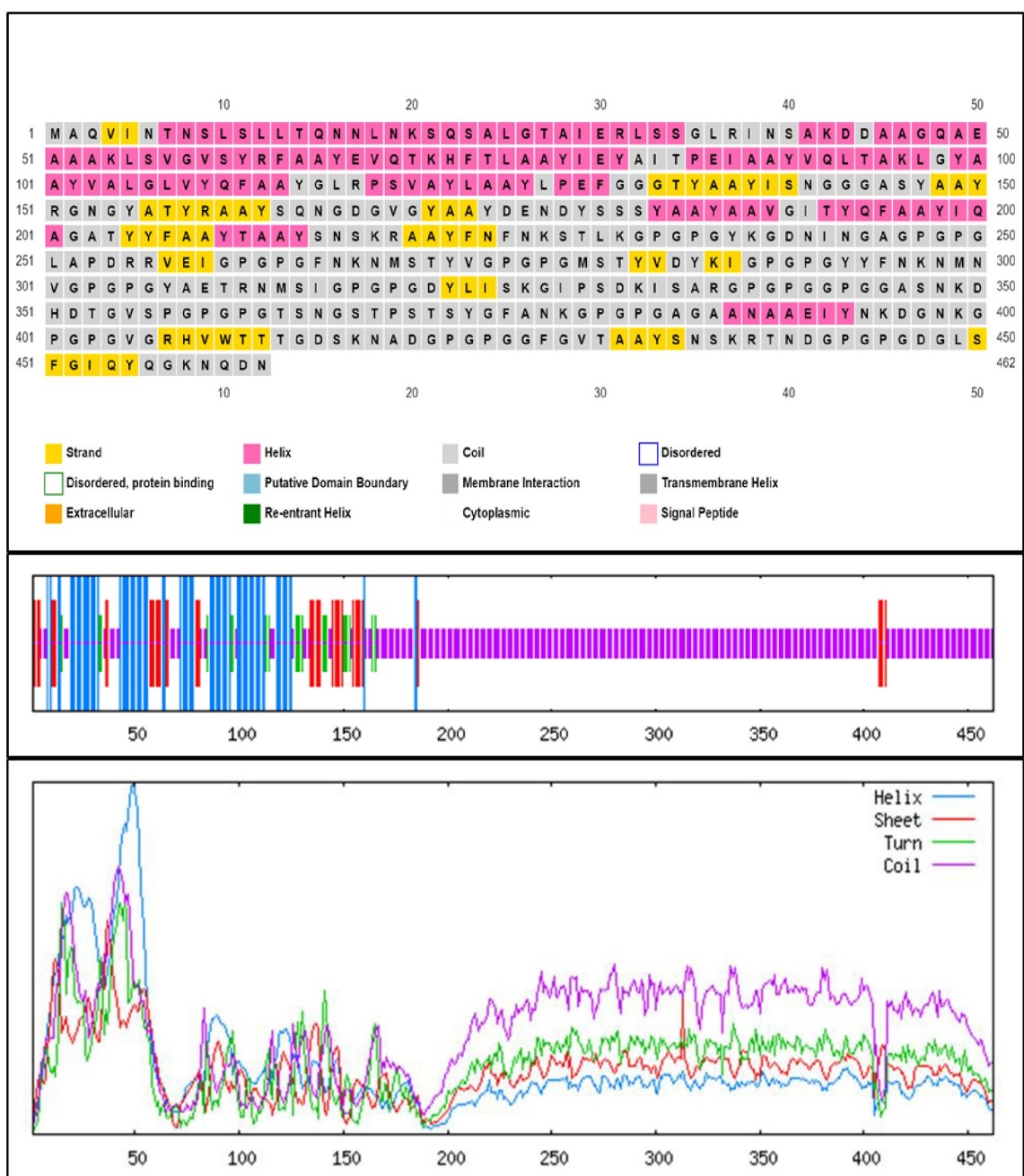

**Fig 5. Graphical representation of secondary structure properties of the MEV using the PSIPRED and SOPMA servers.**

### 3.11 Molecular dynamics simulation and protein-protein binding affinity analysis

Molecular dynamics simulations and protein-protein binding affinity analyses of the TLR-MEV docked complexes were performed using the iMODS tool. This analysis evaluated the stability and physical movements of all TLR-MEV docked complexes, as illustrated in Figs 10–13. The main-chain deformability indicates the ability of a molecule to deform at each of

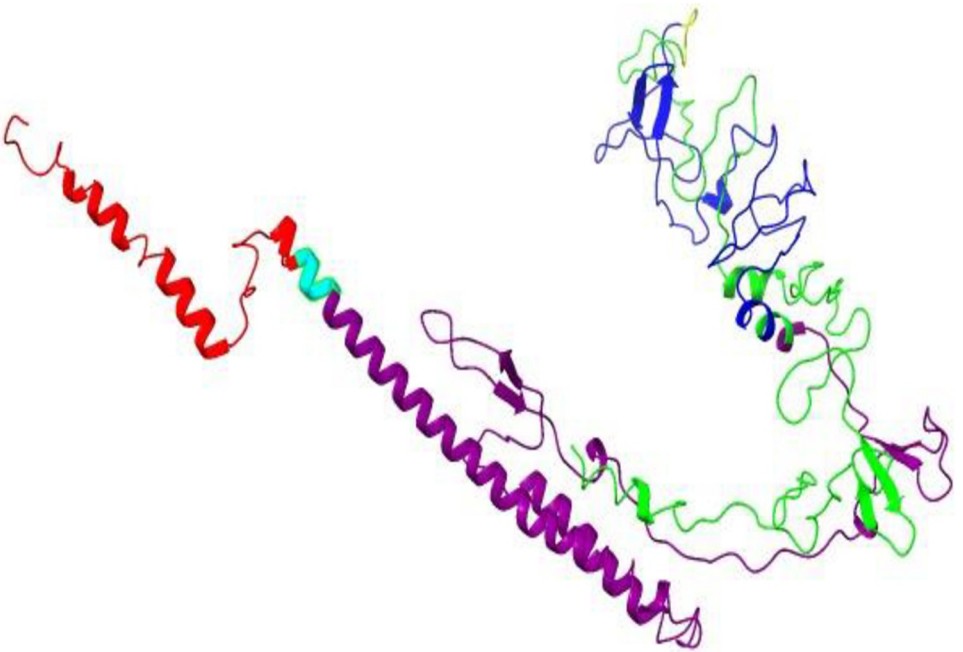

**Fig 6. Tertiary structure of the MEV predicted by I-TASSER and coloured using ChimeraX.** *fliC* (adjuvant) coloured in Red, CTLs in Purple, HTLs in Blue, and B-cell epitopes in Lime-green.

its residues, with regions of high deformability identifying the location of the chain 'hinges'. The B-factor values, calculated by normal mode analysis and proportional to the Root Mean Square, quantify the uncertainty of each atom in the complex.

The eigenvalue of the docked complex represents the energy required to deform the structure, with the following values obtained: $8.6598 \times 10^{-6}$ for TLR1-MEV, $3.6955 \times 10^{-7}$ for TLR2-MEV, $1.6199 \times 10^{-5}$ for TLR4-MEV, and $6.0634 \times 10^{-6}$ for TLR-MEV. The covariance matrix plot shows the correlations between pairs of residues, with red indicating correlated, white indicating uncorrelated, and blue indicating anti-correlated pairs. The elastic network model plot illustrates the connections between atoms, where darker shades of grey represent more rigid connections. For the binding affinity analysis, the Prodigy bioinformatics tool was used, revealing a Pearson's Correlation coefficient (r) of 0.73 (p-value < 0.0001) between the predicted and experimental values, and a Root Mean Square Error (RMSE) of 1.89 kcal mol$^{-1}$. Intermolecular contacts were defined as residues with heavy atoms within 5.5 Å of each other. The Prodigy server predicted 54 intermolecular contacts for the MEV-TLR1 complex, 101 for MEV-TLR2, 168 for MEV-TLR4, and 50 for MEV-TLR5. The binding affinity score (ΔG in kcal mol$^{-1}$) was negative for all docked complexes, as shown in Table 10.

### 3.12 Immune simulation

The administration and repeated exposure to the MEV vaccine resulted in a significant increase in the antibody responses, coupled with a gradual decline in antigen levels over time. A notable Immunoglobulin M (IgM) and Immunoglobulin G (IgG) humoral response indicated some levels of seroconversion. The humoral responses for both IgM and IgG were more pronounced after booster doses compared to the initial dose, with IgM levels being higher than IgG (and its isotypes) (Fig 14a). Following the initial dose, both CD8+ T-cell and CD4

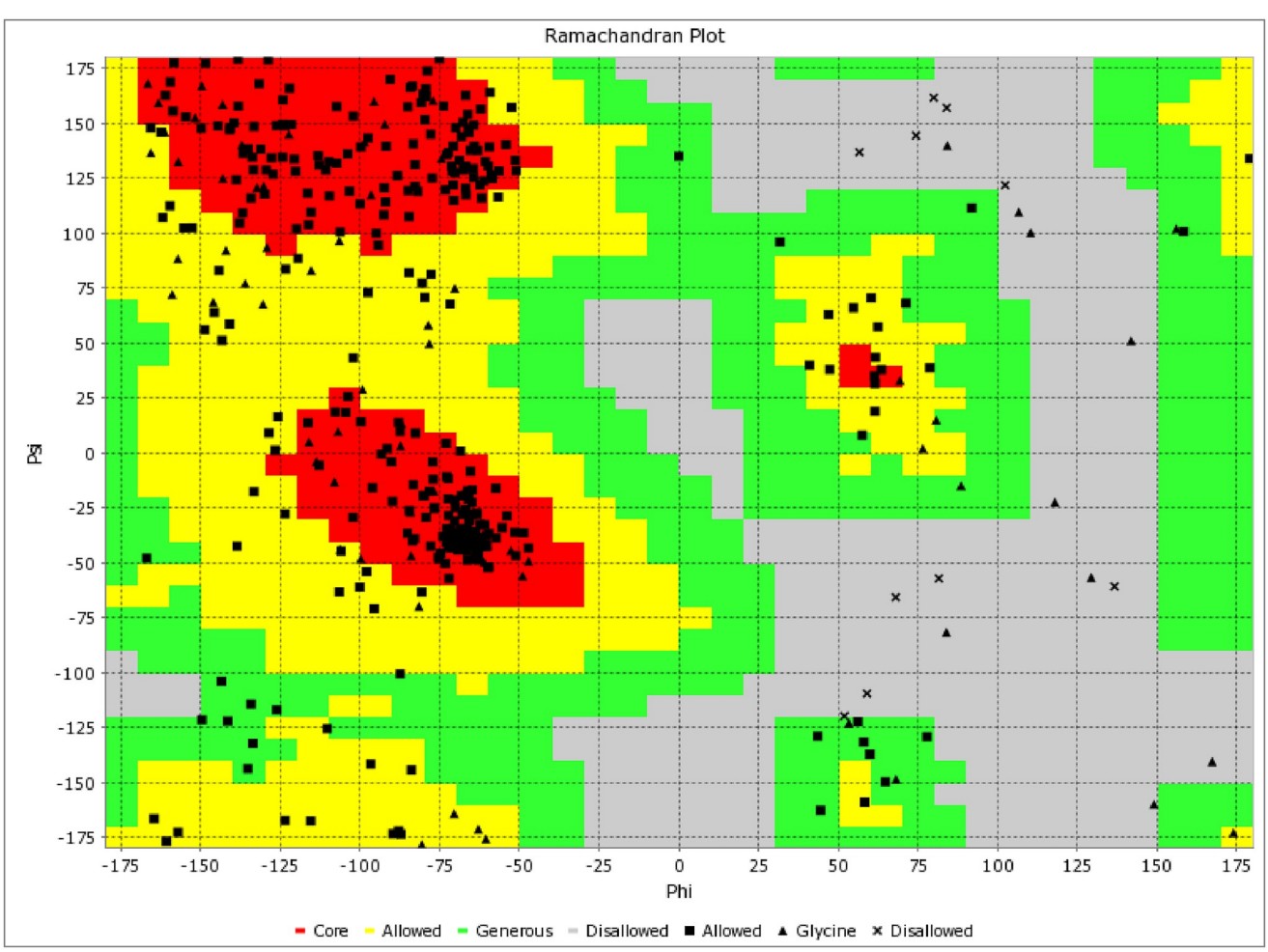

**Fig 7. Validation: Ramachandran plot analysis of protein residues, showing 75.6% in favoured, 21.7% in allowed, and 2.6% in disallowed regions.**

**Table 8. Discontinuous B-cell epitopes of the designed MEV, predicted by ElliPro.**

| S/no | Residues | Number of Residues | Score |
|---|---|---|---|
| 1 | A:M1, A:A2, A:Q3, A:V4, A:I5, A:N6, A:T7, A:N8, A:S9, A:L10, A:S11, A:L12, A:L13, A:T14, A:Q15, A:N16, A:N17, A:L18, A:N19, A:K20, A:S21, A:Q22, A:S23, A:A24, A:L25, A:G26, A:T27, A:A28, A:I29, A:E30, A:R31, A:L32, A:S33, A:S34, A:G35, A:L36, A:R37, A:I38, A:N39, A:S40, A:A41, A:K42, A:D43, A:D44, A:A45, A:A46, A:G47, A:Q48, A:A49, A:E50, A:A51, A:A52, A:A53, A:K54, A:L55, A:S56, A:V57, A:G58, A:V59, A:S60, A:R62 | 62 | 0.869 |
| 2 | A:P235, A:G239, A:N241, A:I242, A:N243, A:G244, A:A245, A:G246, A:P247, A:G248, A:P249, A:P261, A:G262, A:P263, A:G264, A:F265, A:N266, A:K267, A:N268, A:M269, A:S270, A:T271, A:G274, A:P275, A:G276, A:P277, A:G278, A:M279, A:S280, A:Y282, A:V283, A:Y285, A:K286, A:I287, A:G288, A:P289, A:G290, A:P291, A:G292, A:Y294, A:F295, A:N296, A:K297, A:N298, A:M299, A:N300, A:V301, A:G302, A:P303, A:G304, A:P305, A:G306, A:Y307, A:A308, A:E309, A:T310, A:R311, A:N312, A:M313, A:S314, A:I315, A:G316, A:P317, A:G318, A:P319, A:G320, A:D321, A:Y322, A:L323, A:I324, A:S325, A:K326, A:G327, A:I328, A:P329, A:S330, A:D331, A:K332, A:I333, A:S334, A:A335, A:R336, A:G337, A:P338, A:P343, A:G344, A:G345, A:A346, A:S347, A:N348, A:K349, A:D350, A:H351, A:D352, A:T353, A:G354 | 96 | 0.681 |
| 3 | A:A88, A:V91, A:Q92, A:T94, A:A95, A:K96, A:L97, A:G98, A:Y99, A:A100, A:A101, A:Y102, A:V103, A:A104, A:L105, A:G106, A:L107, A:V108, A:Y109, A:Q110, A:F111, A:A112, A:A113, A:Y114, A:G115, A:L116, A:R117, A:P118, A:S119, A:V120, A:Y122, A:L123, A:D175, A:E176, A:N177, A:D178, A:Y179, A:S180, A:S181, A:S182, A:Y183, A:A184, A:A185, A:Y186, A:A187, A:A188, A:V189, A:G190, A:I191, A:T192, A:Y193, A:Q194, A:F195, A:A196, A:A197, A:Y198, A:I199, A:Q200, A:A201, A:G202, A:A203, A:T204, A:Y205, A:Y206, A:F207, A:A208, A:H408, A:V409, A:W410, A:T411, A:T412, A:T413, A:G414, A:D415, A:S416, A:K417, A:N418, A:A419, A:D420, A:G421, A:P422, A:G423, A:P424, A:G425, A:G426, A:F427, A:G428, A:V429, A:T430, A:A431, A:A432, A:Y433, A:S436, A:R438, A:N440, A:G442, A:P443, A:D447, A:G448, A:Q456 | 100 | 0.632 |

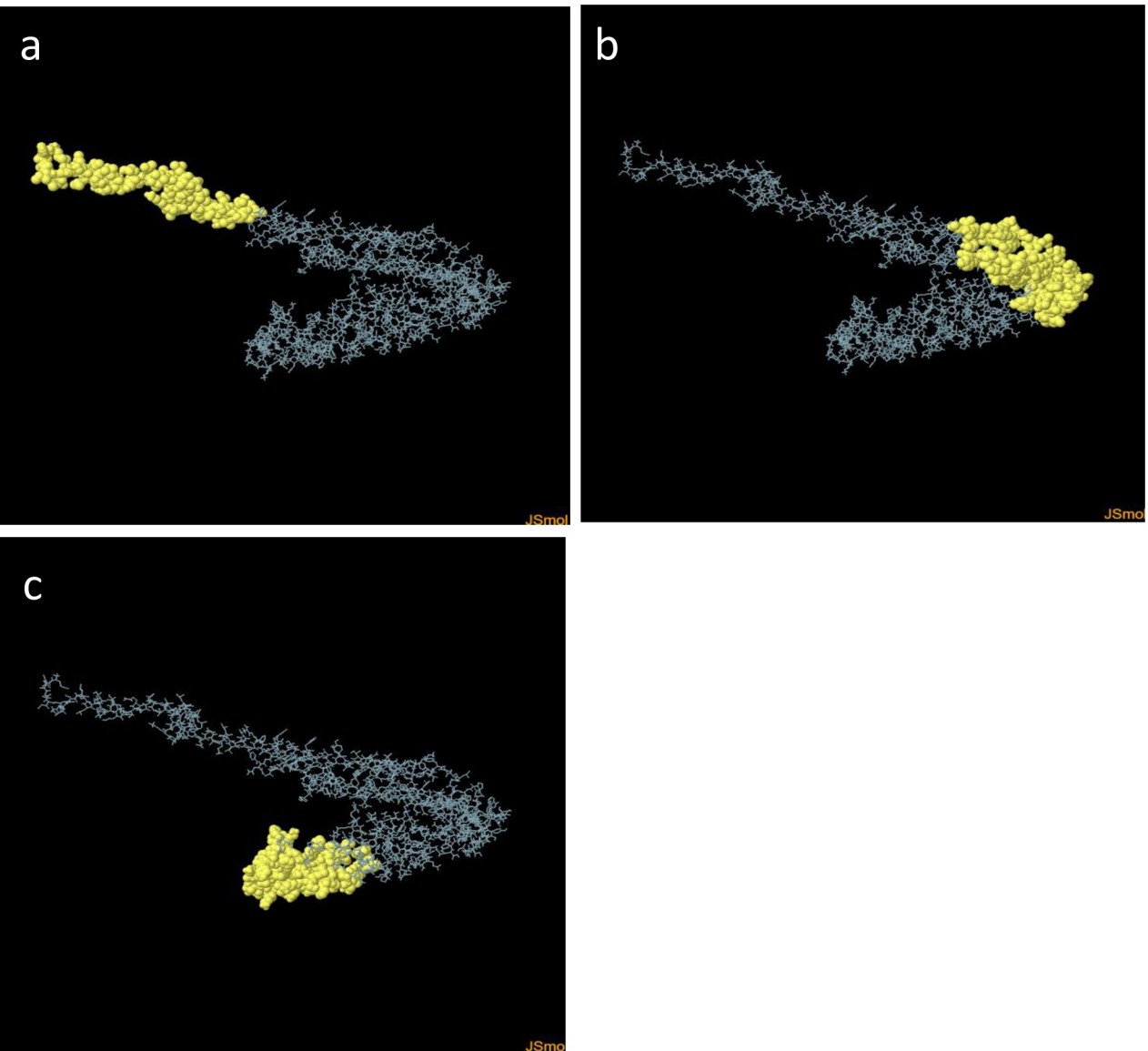

**Fig 8. 3D representation of the discontinuous B-cell epitopes of MEV.** The yellow surface shows discontinuous B-cell epitopes, while the grey stick represents the bulk of the polyprotein. (a) With 61 residues and a score of 0.869, (b) with 96 residues with a score of 0.681, and (c) with 100 number of residues with a score of 0.632.

**Table 9. Result of docking 3D MEV model with TLRs using ClusPro 2.0 server.**

| Protein-Protein Dock complex | Best model (Cluster) | Center Energy | Lowest Energy |
|---|---|---|---|
| MEV-TLR1 | 8 | -1112.4 | -1220.2 |
| MEV-TLR2 | 7 | -1396.7 | -1396.7 |
| MEV-TLR4 | 1 | -1002.2 | -1213.1 |
| MEV-TLR5 | 2 | -1631.5 | -1631.5 |

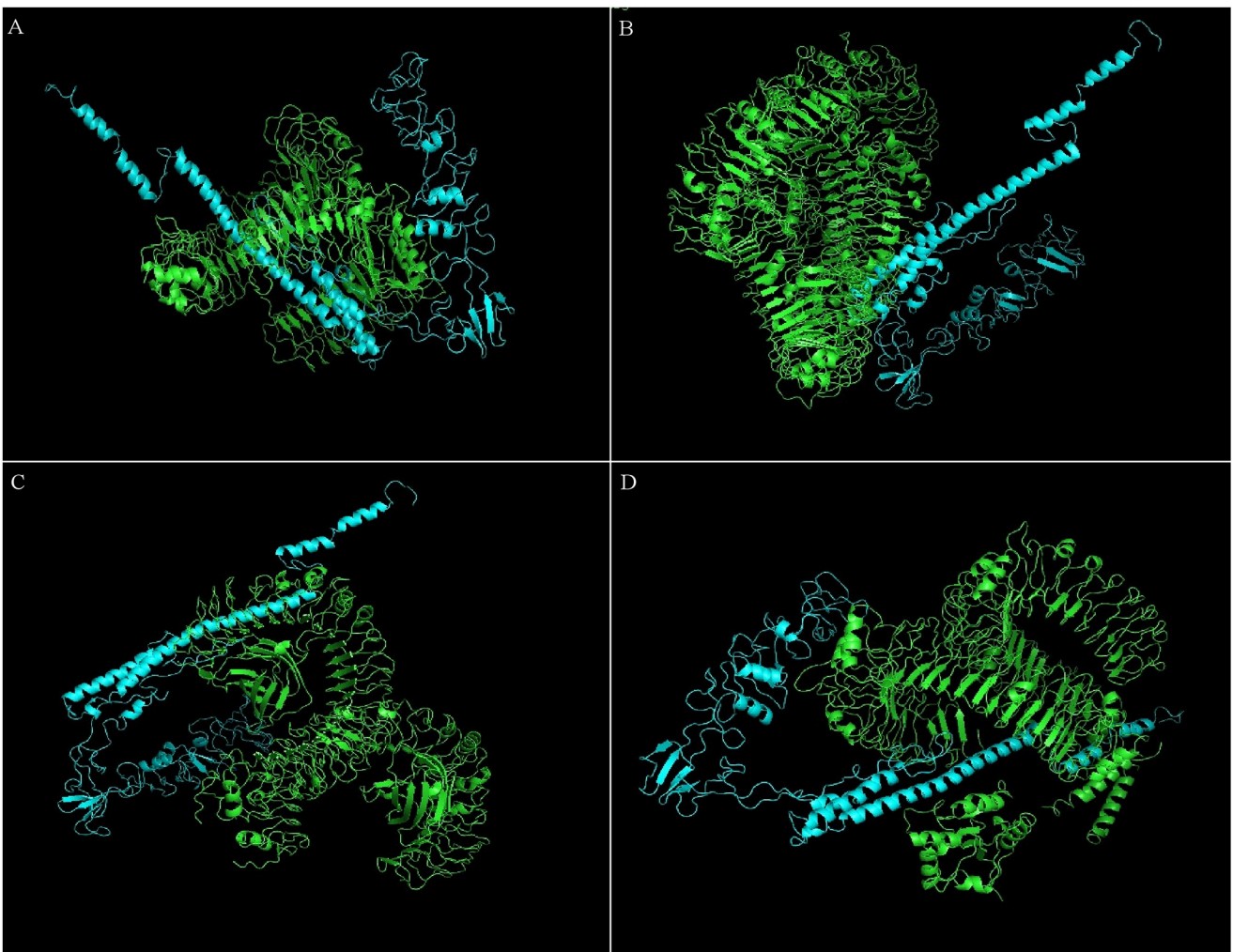

**Fig 9. Docking of the MEV 3D model (cyan) with TLRs (green): (a) MEV construct with TLR-1 (PDB ID: 6NIH), (b) MEV construct with TLR-2 (PDB ID: 6NIG), (c) MEV construct with TLR-4 (PDB ID: 3FXI), and (d) MEV construct with TLR-5 (PDB ID: 3JOA).** All models were visualized and colored with PyMOL 3.0.4 software.

+ responses were actively stimulated. CD4+ T-helper lymphocytes peaked after the second booster dose, with a significant increase in the active entity state and T-helper-memory (Th) phenotype observed after the first booster, remaining stable with minimal decay over 100 days (Fig 15a and 15b). Similarly, the prime dose induced a significant spike in the IFN-γ response (associated with both CD8+ T-cell and CD4+ Th1 response) and a substantial increase in IL-2, IL-10, IL-12, and Transforming Growth Factor-beta (TGF-b) cytokine responses, linked to the regulatory T-cells phenotype (TGF-b and IL-10). These responses persisted after the two booster doses, with IFN-g response maintained and IL-2, IL-10, IL-12, and TGF-b levels rising significantly (Fig 16b). These immune readouts are typically associated with a broader peak in antigen quantification.

Analysis of the B-cell population showed that overall B-cell counts and B-cell memory responses were higher and stable, with minimal decay. B-cell memory peaked after the second booster dose and remained stable over 100 days. Similarly, the active B-cell population was higher and stable over the same period (Fig 14b and 14c). There was also an increase in the

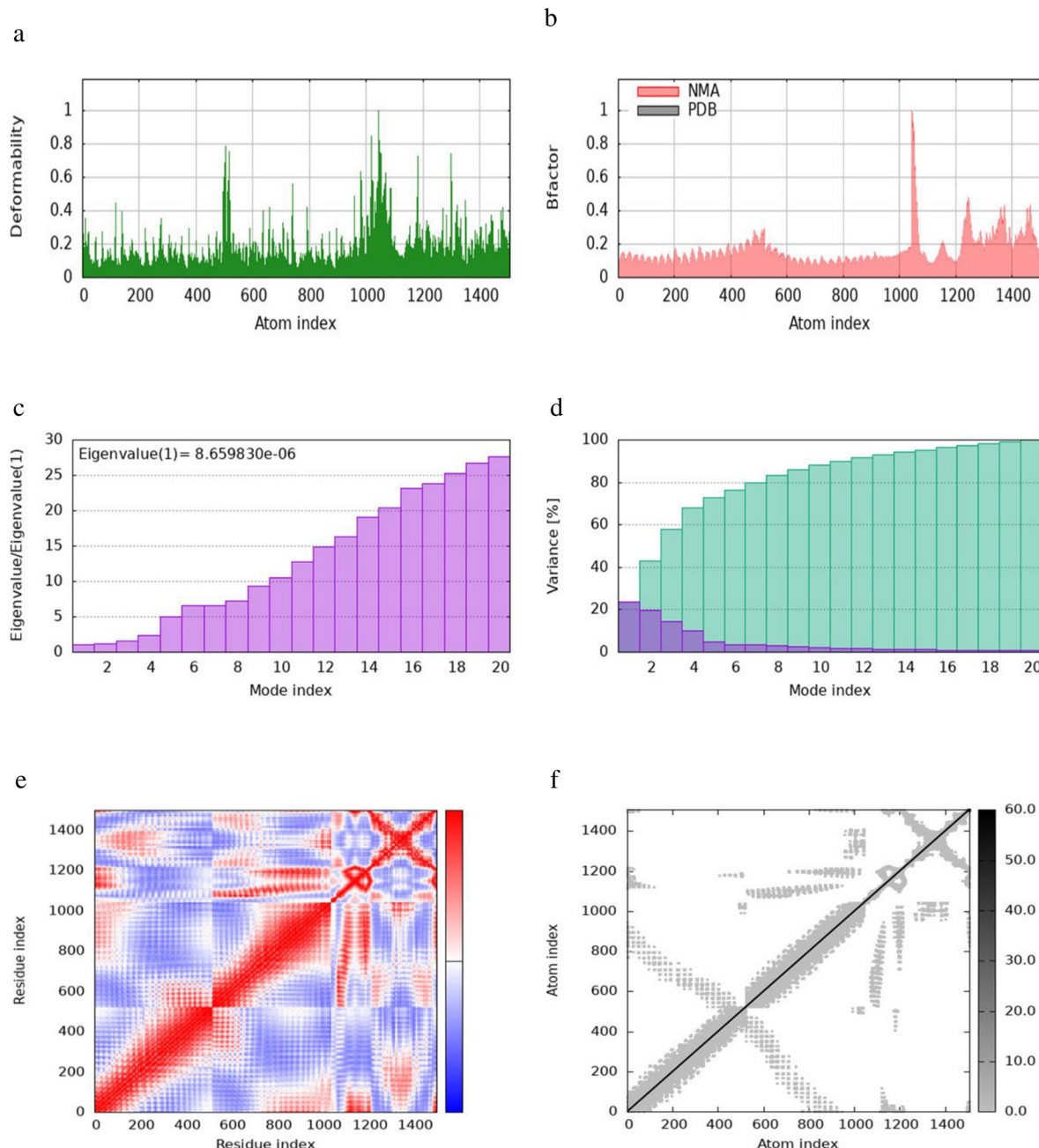

**Fig 10. iMODS simulation of the TLR1-MEV docked complex.** (a) Main-chain deformability reveals residue flexibility. (b) B-factor values indicate the positional uncertainty of each atom. (c) Eigenvalues represent the energy required for deformation. (d) Variance per normal mode is inversely proportional to eigenvalue, with individual (purple) and cumulative (green) variances. (e) The covariance matrix shows residue pair interactions. (f) The elastic network model depicts atom connections with darker shades indicating stiffer springs.

activity of dendritic cells, natural killer (NK) cells, and macrophages throughout the simulation (Fig 17b–17d). This indicates the vaccine's effectiveness in stimulating the appropriate immunological components for an effective response. Overall, these results demonstrated a steady and concomitant increase in immune responses over time with each vaccination schedule.

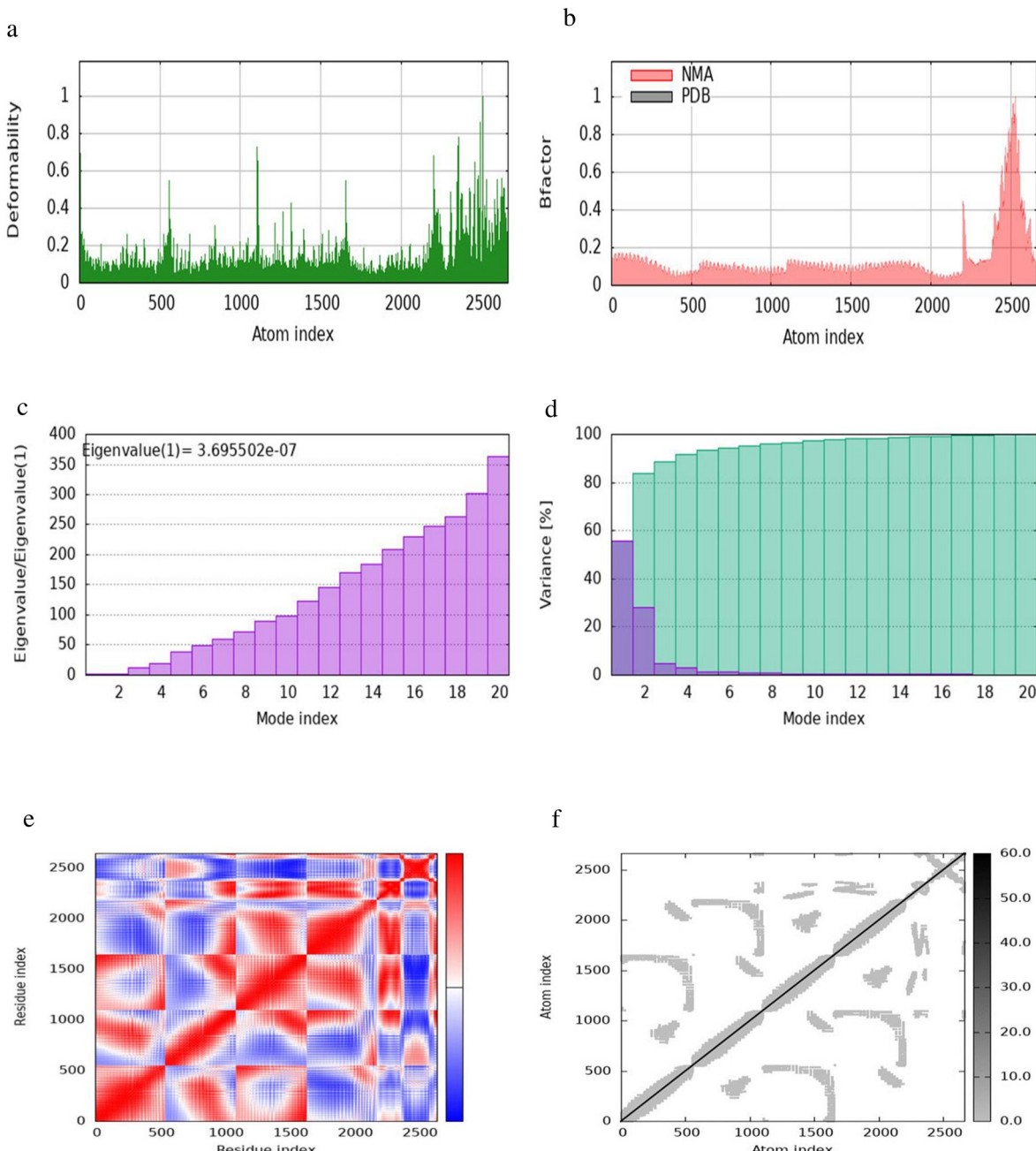

**Fig 11. iMODS simulation of the TLR2-MEV docked complex.** (a) Main-chain deformability reveals residue flexibility. (b) B-factor values indicate the positional uncertainty of each atom. (c) Eigenvalues represent the energy required for deformation. (d) Variance per normal mode is inversely proportional to eigenvalue, with individual (purple) and cumulative (green) variances. (e) The covariance matrix shows residue pair interactions. (f) The elastic network model depicts atom connections with darker shades indicating stiffer springs.

### 3.13 Codon adaptation and *In-Silico* plasmid cloning

The optimized vaccine codon sequence comprised 1,386 nucleotides. Initially, the GC content was 40.89%, but it increased to 70.78% after optimization with JCAT, enhancing translational efficiency. The codon adaptation index was calculated to be 0.956. Restriction enzyme sites on the JCAT-optimized MEV sequence were identified using NEBcutter Version 3.0.17 software.

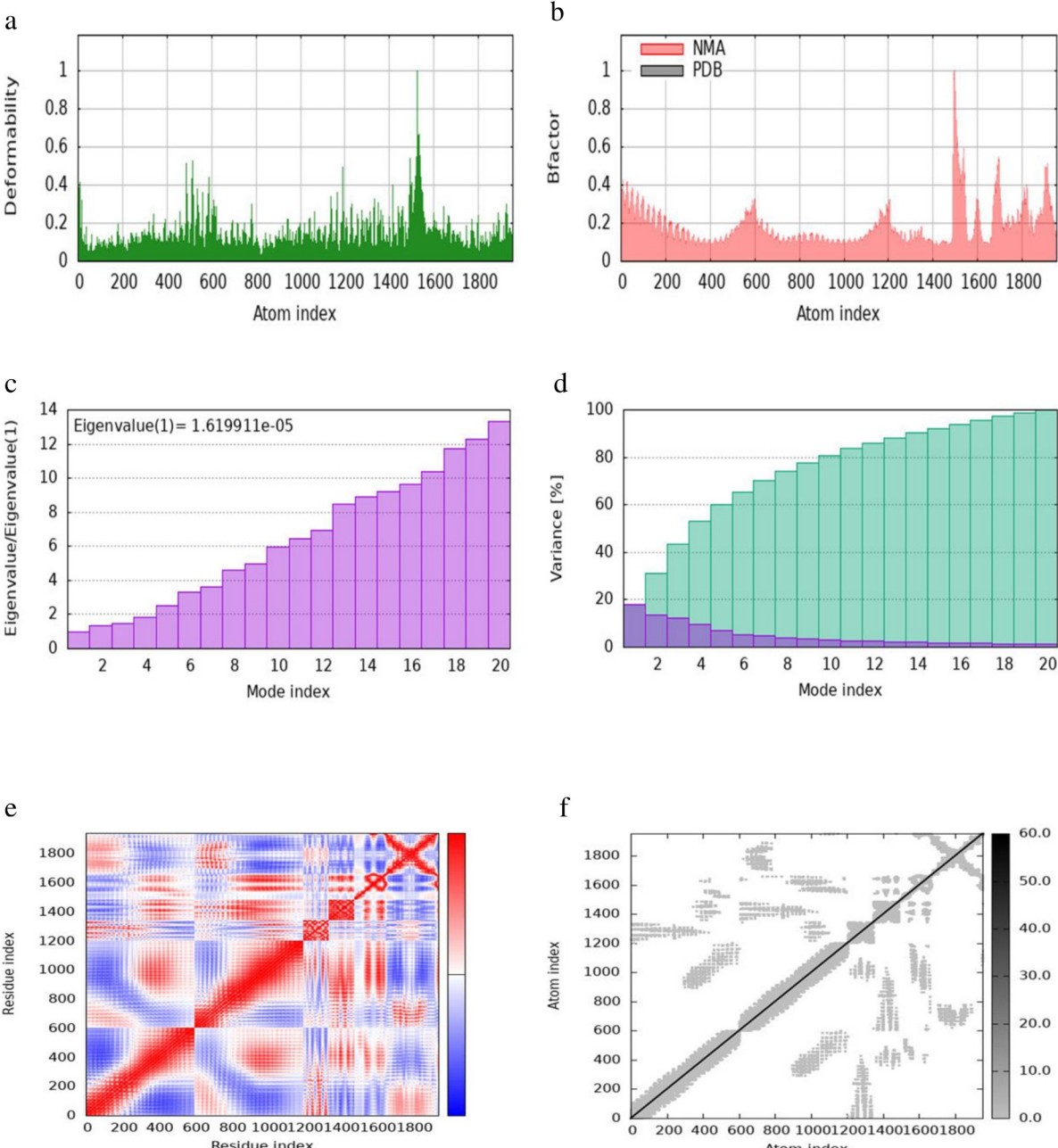

**Fig 12. iMODS simulation of the TLR4-MEV docked complex.** (a) Main-chain deformability reveals residue flexibility. (b) B-factor values indicate the positional uncertainty of each atom. (c) Eigenvalues represent the energy required for deformation. (d) Variance per normal mode is inversely proportional to eigenvalue, with individual (purple) and cumulative (green) variances. (e) The covariance matrix shows residue pair interactions. (f) The elastic network model depicts atom connections with darker shades indicating stiffer springs.

The MEV coding DNA was inserted into the multiple cloning sites between HindIII and BamHI. *In-silico* cloning, performed with SnapGene software, generated a Plasmid-MEV clone with a total length of 6844 bp (Fig 18).

The red segment represents the MEV coding gene, while the black segment indicates the plasmid vector backbone. The plasmid DNA size is 6844 bp.

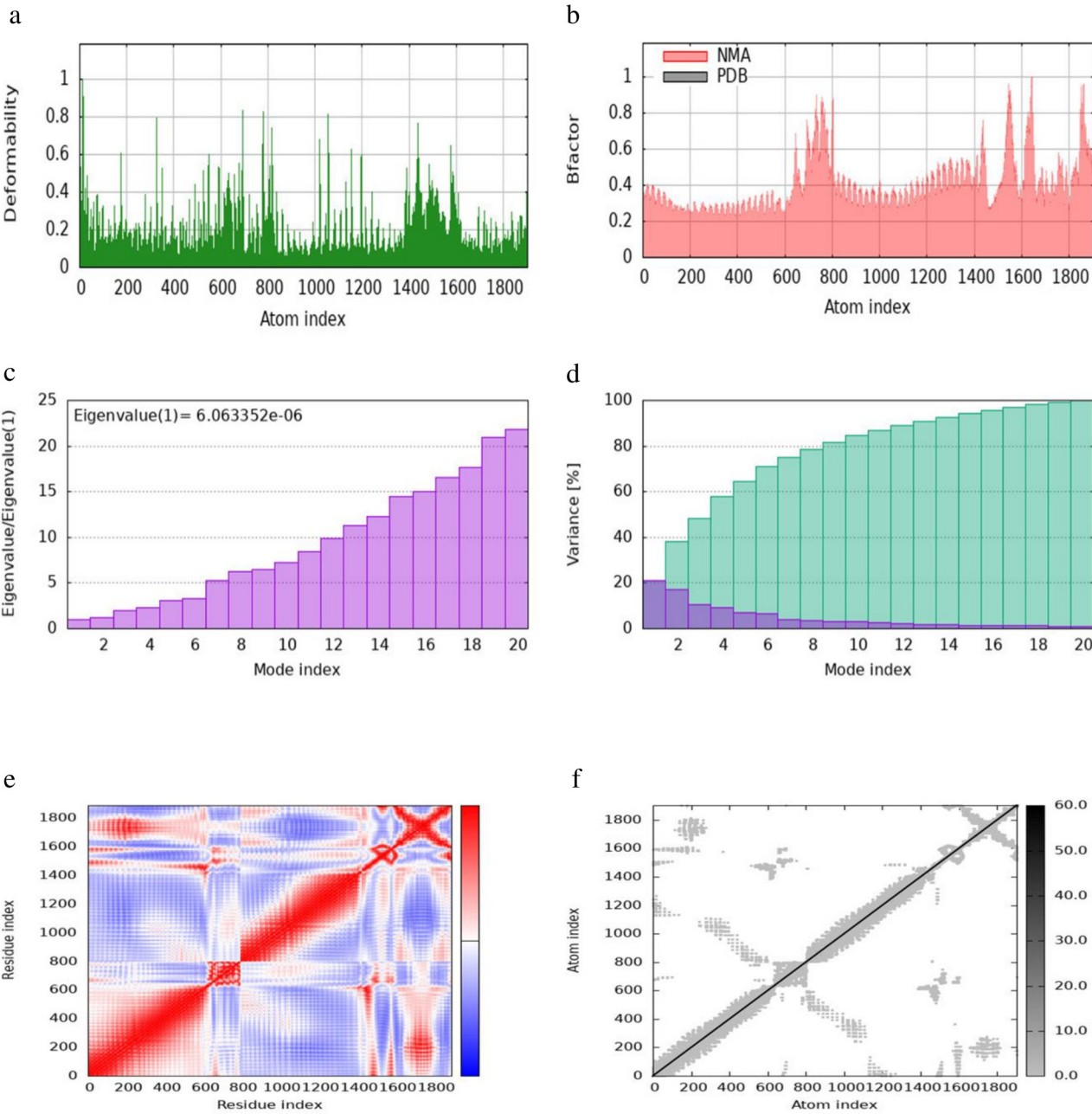

**Fig 13. iMODS simulation of the TLR5-MEV docked complex.** (a) Main-chain deformability reveals residue flexibility. (b) B-factor values indicate the positional uncertainty of each atom. (c) Eigenvalues represent the energy required for deformation. (d) Variance per normal mode is inversely proportional to eigenvalue, with individual (purple) and cumulative (green) variances. (e) The covariance matrix shows residue pair interactions. (f) The elastic network model depicts atom connections with darker shades indicating stiffer springs.

## 4.0 Discussion

OMPs play crucial roles in antibiotic resistance by acting as gatekeepers that regulate the influx of external molecules. These proteins are also pivotal in cellular communication and have promising applications as biosensors. Their multi-functionality makes them excellent

**Table 10. Prediction of binding affinity and dissociation constant between the docked complexes.**

| Protein-protein complex | ΔG (kcal mol⁻¹) | $K_d$ (M) at 25°C | ICs charged-charged | ICs charged-polar | ICs charged-apolar | ICs polar-polar | ICs polar-apolar | ICs apolar-apolar | NIS charged | NIS apolar |
|---|---|---|---|---|---|---|---|---|---|---|
| MEV-TLR1 | -10.4 | 2.3e-08 | 13 | 20 | 9 | 3 | 7 | 2 | 27.61 | 26.01 |
| MEV-TLR2 | -9.4 | 1.4e-07 | 12 | 12 | 13 | 15 | 14 | 35 | 28.08 | 28.83 |
| MEV-TLR4 | -18.1 | 5e-14 | 23 | 33 | 38 | 13 | 36 | 25 | 24.81 | 32.01 |
| MEV-TLR5 | -8.8 | 3.6e-07 | 3 | 6 | 11 | 0 | 9 | 21 | 19.56 | 42.13 |

Key:

**ICs** –Intermolecular Contacts

**ΔG** –Binding Affinity (Gibbs free energy)

**$K_d$** –Dissociation constant

**NIS**–Non-interacting surface

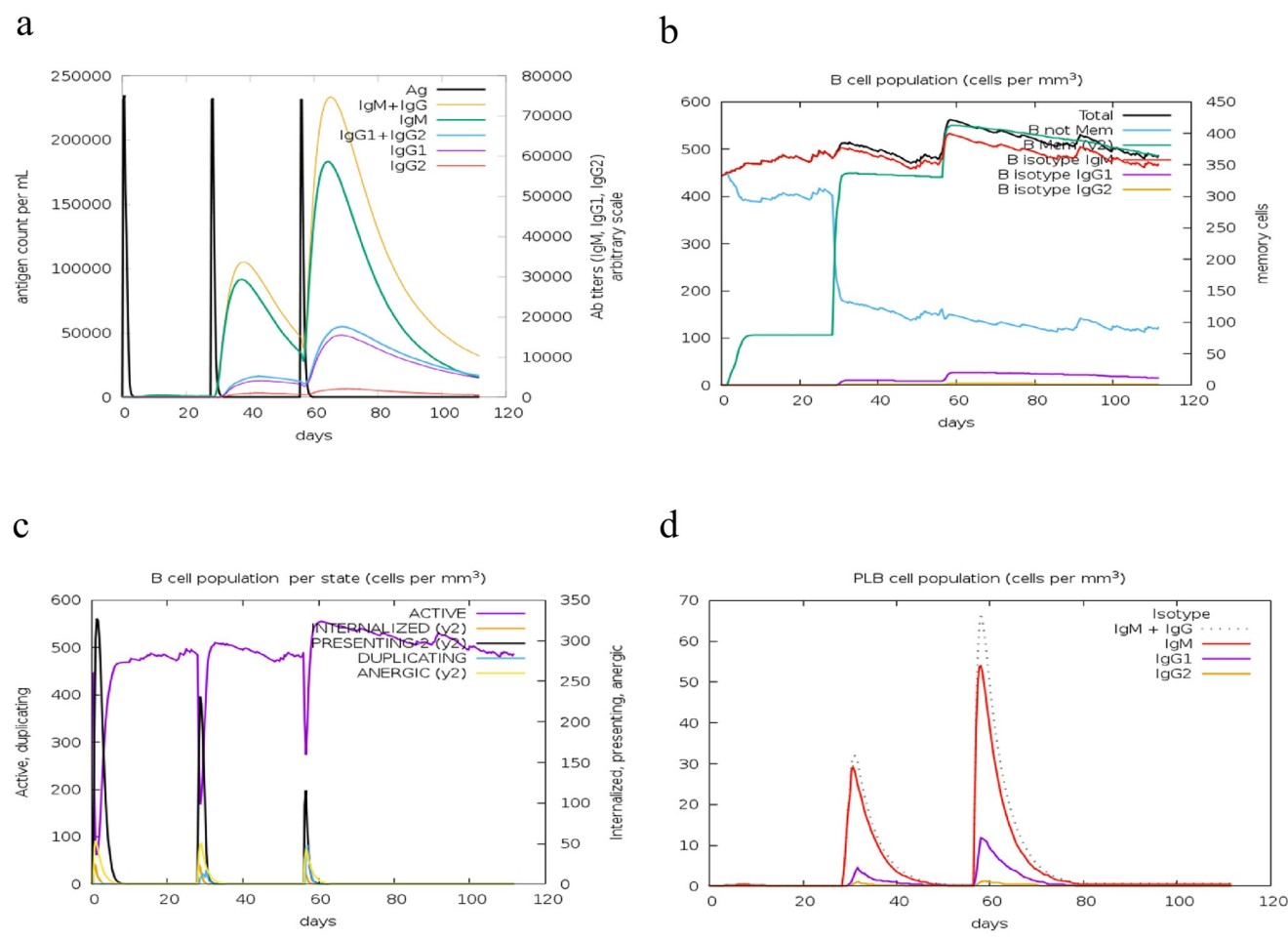

**Fig 14. Distribution of B-lymphocytes and their isotypes.** (a) Antigen and immunoglobulin levels, with antibodies classified by isotype. (b) Total B-lymphocyte count, memory cells, and isotypes IgM, IgG1, and IgG. (c) B-lymphocyte population by entity-state, showing counts for active, class-II presenting, antigen-internalized, duplicating, and anergic states. (d) Plasma B lymphocyte (PLB) count divided by isotype (IgM, IgG1, and IgG2).

a

b

c

d

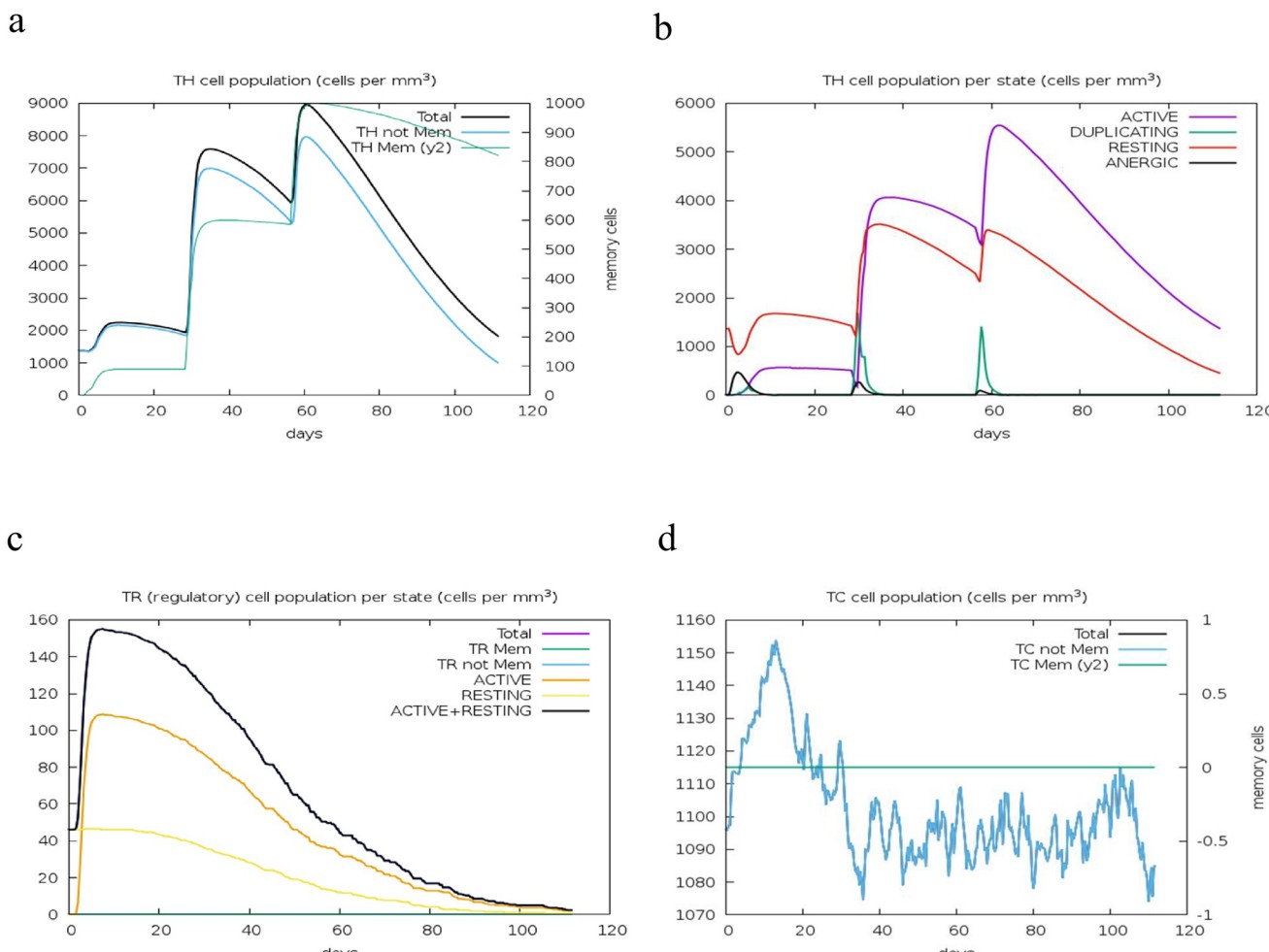

**Fig 15. T-lymphocyte populations.** (a) CD4+ T-helper lymphocyte counts, showing total and memory cells. (b) CD4+ T-helper lymphocyte counts by entity state, including active, resting, anergic, and duplicating states. (c) CD4+ T-regulatory lymphocyte counts, with total, memory, and entity-state counts plotted. (d) CD8+ T-cytotoxic lymphocyte counts, showing total and memory cells.

candidates for vaccine development. This study leverages the immunogenic properties of OMPs, along with the *fliC*, to design an innovative MEV. The computational approach to designing chimeric membrane proteins aims to optimize the advantages of these molecular gateways and signaling networks [72–74].

In our vaccine study, the selected OMP sequences represented highly conserved regions across different OMP classes (Table 2). These conserved regions, consistent across various strains or species, offer a promising foundation for developing vaccines with broad-spectrum efficacy. For instance, targeting conserved regions in influenza viruses has proven effective in developing universal flu vaccines [75]. Additionally, because these regions are crucial for the pathogen's survival, they are less prone to mutations. This stability reduces the likelihood that the pathogen will evolve mechanisms to evade the immune response, enhancing the vaccine's long-term effectiveness [75–77].

Through phylogenetic analysis, we identified distinct levels of diversity among OmpA, OmpC, and OmpF (see Fig 2), which reflect a broader spectrum of antigenic variants. By

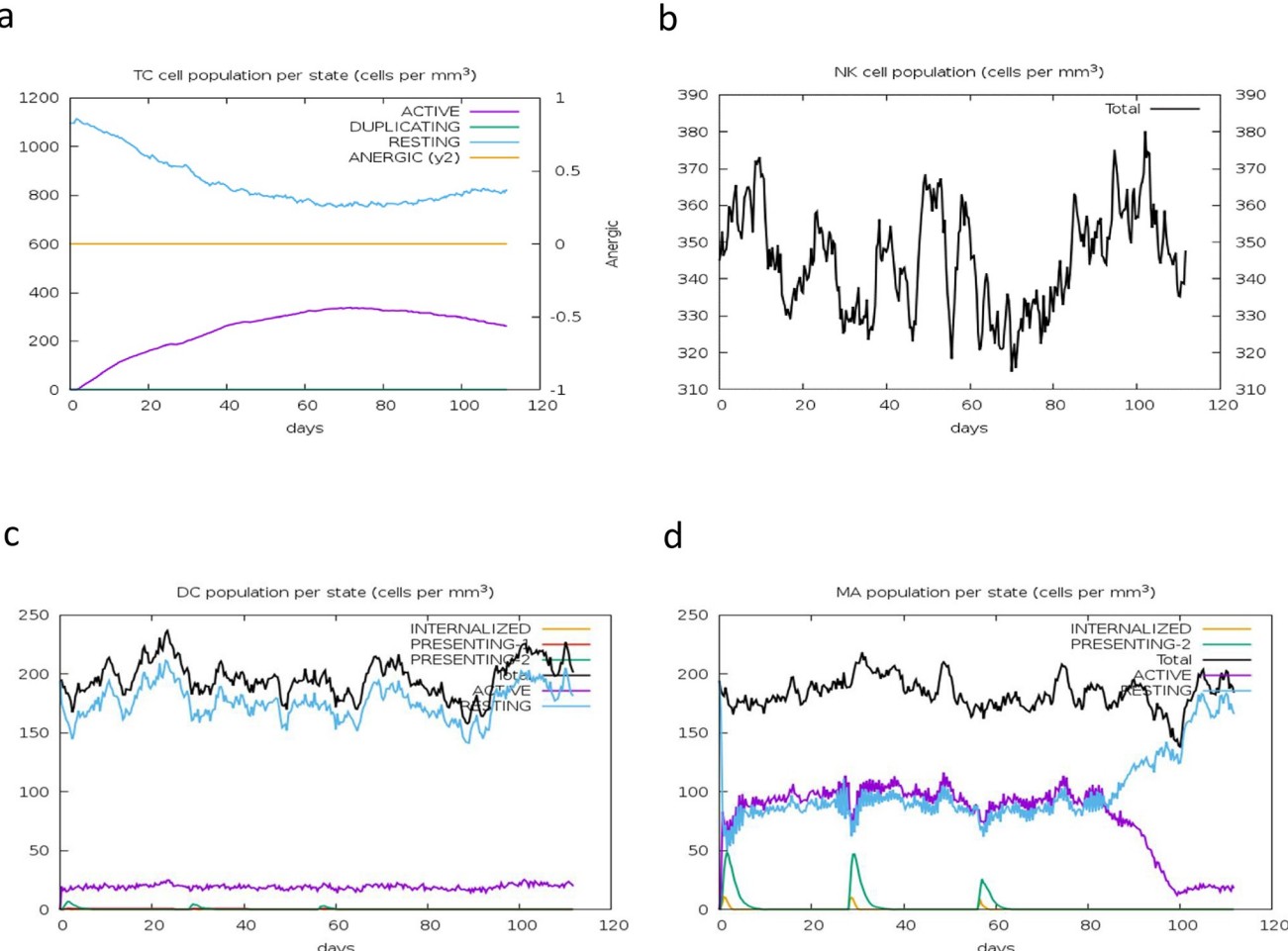

**Fig 16. Epithelial cells and cytokines.** (a) epithelial cell counts, broken down into active, antigen-infected, and class-I MHC presenting states. (b) The concentration of cytokines and interleukins, with the inset plot showing danger signal (D) and leukocyte growth factor IL-2.

integrating phylogenetic insights and conservancy, our study selected a strategic combination of OMPs that optimizes the immunogenic response while minimizing redundancy, ultimately enhancing the overall effectiveness and efficiency of the vaccine [9, 10, 78]

This study identified 29 epitopes that could stimulate both cellular and humoral immune responses, targeting both T-cells and B-cells. Specifically, across the three classes of OMPs, 15 CTL, 8 HTL, and 7 linear B-cell epitopes were predicted based on predefined thresholds. Notably, the epitope 'IEYAITPEI' was identified in both MHC-I and MHC-II epitopes of OmpA, as well as in MHC-I epitopes of OmpC. All epitopes were highly conserved with a minimum and maximum conservancy identity at 100% (Table 4), ensuring the vaccine is efficacious against the diverse bacterial variants. All the predicted epitopes were non-allergenic and non-toxic, therefore reducing the likelihood of the vaccine candidates causing type-II hypersensitivity reactions or cellular apoptosis due to toxic reactions. Furthermore, the highly exposed residues of CTL epitopes are crucial in eliciting a robust cytotoxic response, as CTLs recognize pathogen antigens presented by MHC-I molecules, which trigger these responses [79]. HTL epitopes, on the other hand, are instrumental in inducing the release of IFN-γ, making both CTL and HTL epitopes essential components of our MEV design.

a

b

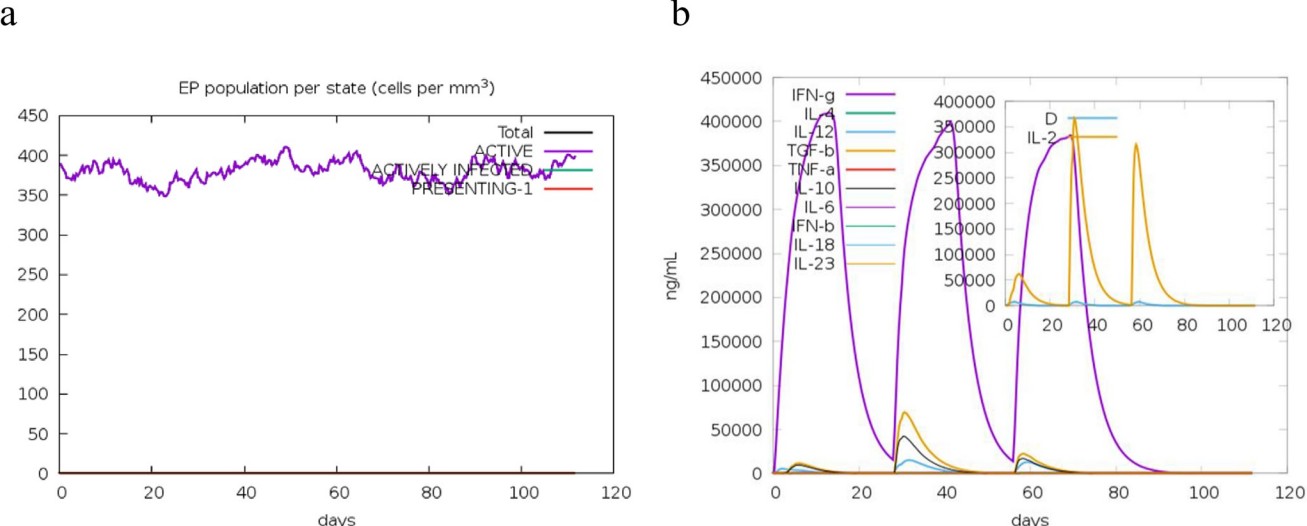

**Fig 17. Immune cell activity.** (a) CD8+ T-cytotoxic lymphocyte counts by entity-state. (b) Total count of natural killer cells. (c) Dendritic cells, which present antigenic peptides on both MHC class-I and class-II molecules, with the total number broken down into active, resting, internalized, and antigen-presenting states. (d) Macrophage counts, including total, internalized, class-II presenting, active, and resting states.

IFN-γ, a critical cytokine released by CD4+ T-cells, is essential for activating effector cells and facilitating antibody-mediated responses to infections [33, 35]. The 8 HTL epitopes predicted across the three classes of OMPs were further evaluated for their ability to induce IFN-γ. Our study identified HTL epitopes, 'LAPDRRVEI' and 'IEYAITPEI', as potential inducers of IFN-γ production (Table 6). While only two epitopes were predicted to induce IFN-γ, this result is considered satisfactory. IFN-γ is mainly expected to initiate a cascade of receptor-ligand interactions involving other cytokines and pattern recognition receptors (PRRs), such as IL-4, Tumor Necrosis Factor-alpha (TNF-α), and lipopolysaccharides [33, 80], as excessive levels of IFN-γ can lead to significant tissue damage, necrosis, and inflammation, potentially worsening disease pathology [33]. The ability of our HTL epitopes to induce IFN-γ is particularly important for mediating intestinal immunity against *Salmonella spp*. [33, 81].

Seven linear B-cell epitopes were identified as antigenic based on the antigenic potential of their hydrophobic residues, such as cysteine, leucine, and valine, which are often present on protein surfaces [82]. Details on their positions, lengths, sequences, and antigenicity are provided in Table 5. These epitopes were selected using an artificial neural network and the Kolaskar and Tongaonkar antigenicity scale from the IEDB server, which has a 75% accuracy in predicting antigenic determinants of proteins. The identified B-cell epitopes were non-toxic and non-allergenic, making them suitable candidates for vaccine development.

When designing a vaccine, it is important to consider the population coverage capacities of the predicted MHC-I and MHC-II T-cell epitopes. Given the presence of over a thousand human MHC alleles worldwide, it is essential to ensure that the vaccine can elicit an immune response in individuals with diverse MHC alleles [83, 84]. Our vaccine demonstrated a global population coverage of 94.91%, indicating that approximately 95% of the global population would likely be protected if vaccinated with our construct containing the identified CD8+ and CD4+ T-cell epitopes. Europe showed the highest coverage, at 97.52% (Fig 3), suggesting significantly higher protection from our vaccine chimera in this region compared to others. Given that the *S*. Kentucky ST198 MDR clone was first identified in Egypt, and is now

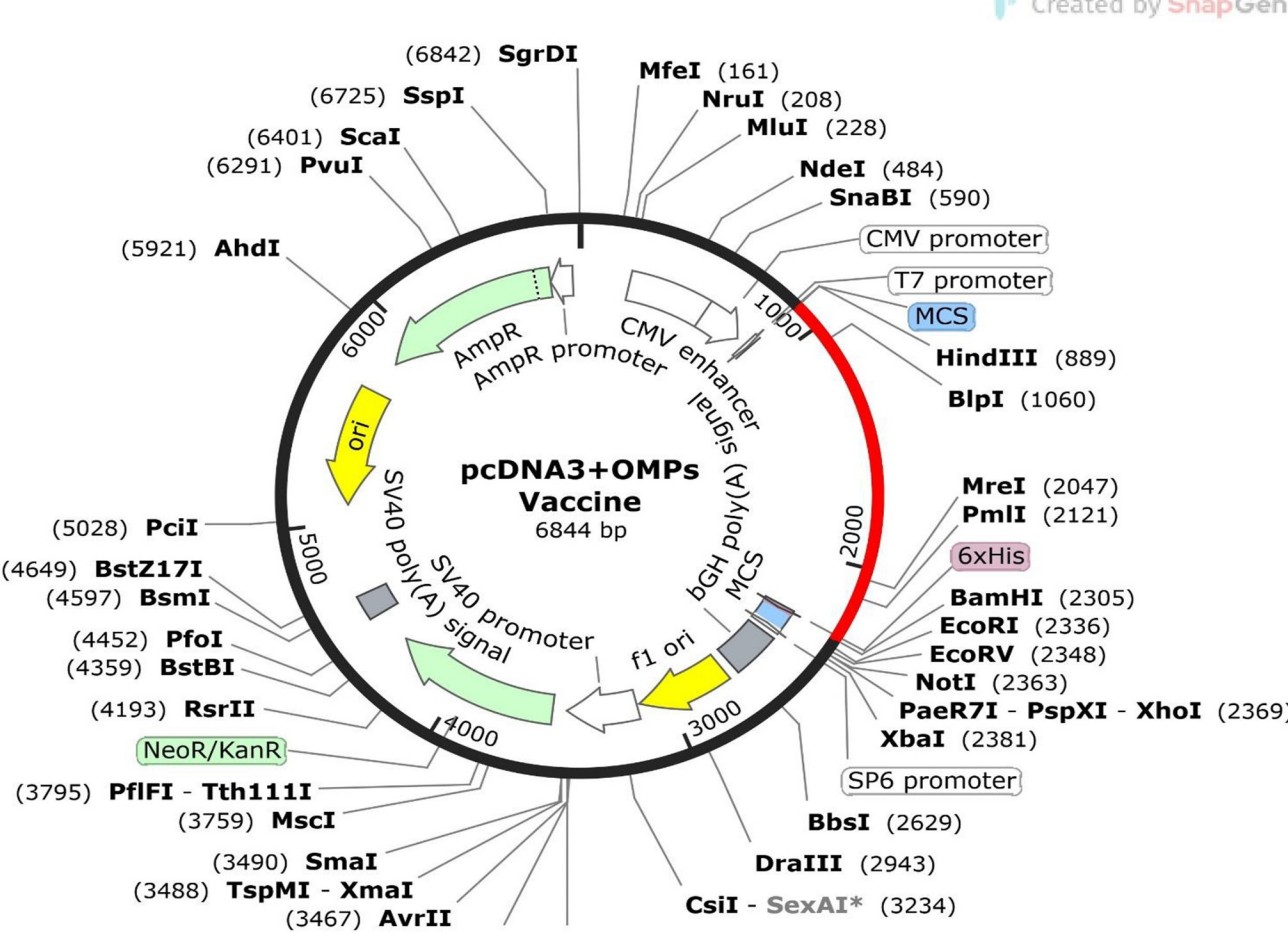

**Fig 18.** *In-Silico* **cloning of the final MEV in pcDNA3.1_CT-GFP expression vector.**

widespread in Africa [3, 85–87], we were particularly interested in the vaccine's performance across different African regions. The vaccine coverage in Africa was substantial at 85.14%, with CD8+ T-cell epitopes achieving higher coverage than CD4+ T-cell epitopes (Table 7). North Africa exhibited the highest coverage on the continent, with a combined average of 92.89% for both classes. The CD4+ T-cell coverage in North Africa was notably higher at 52.24%, nearly double that of other African regions for this T-cell class (Table 7). The variations in regional coverage may be attributed to differences in ethnicity and race. North Africa's mixed population, which includes Arabs, Caucasoids, Blacks, and Tunisian Berbers, shares more allele similarities with Europe and sub-Saharan Africa compared to other parts of Africa. Discrepancies in population coverage data could arise from the IEDB server's allele database, which has more entries from Europe and the Americas than from Africa and Asia [39]. This disparity may contribute to the variations observed in our study.

The MEV was constructed by integrating predicted T-cell epitopes, B-cell epitopes, and the *fliC*, interconnected using AAY, GPGPG, and EAAK protein linkers respectively (Figs 4 and 6). These linkers are crucial for preserving the immunogenicity and properties of proteins within the MEV chimera. The absence of linkers in MEV construction could lead to the

formation of a novel protein with unknown characteristics or the generation of neo-epitopes and junctional epitopes [88, 89]. From an immunological perspective, protein fusions are advantageous as each antigen in the fusion can benefit from additional T-cell help provided by partner T-cell epitopes, resulting in enhanced antigen-specific immunogenicity and antibody titers [90]. The physicochemical analysis of the MEV construct indicated a highly antigenic, non-allergenic, and non-toxic chimera with a molecular weight of 47.343 kD and a theoretical pI of 9.150. The molecular weight of peptides used in MEV should ideally be less than 110 kD, and the theoretical pI score suggests a basic nature of our MEV chimera [91, 92]. The MEV was predicted to be thermo-stable; the computed aliphatic index of the protein was 57.23. Proteins with an aliphatic index of over 50 are associated with enhanced thermo-stability [93]. The physicochemical analysis also showed that the MEV is soluble (hydrophilic), and predominantly composed of random coils (69.91%) in its protein secondary structure (Fig 5). The abundance of random coils in the MEV underscores its potential for efficient antigenicity and may also reflect the natural antigenic nature of the native unfolded alpha-helices secondary structure [18, 92, 94]. Analysis using the Ramachandran plot showed that over 95% of the residues in the MEV model were within the allowed regions (Fig 7), indicating a satisfactory 3D model. ERRAT analysis further confirmed the stereo-chemical accuracy of the MEV's 3D structure.

The evaluation of protein-protein interactions revealed that the least energy values of all our docked complexes indicate a stable TLR-MEV interaction (see Table 9). Molecular simulation analysis further suggested a low probability of complex deformation during immune response activity, as evidenced by their maximum eigenvalues (Figs 10–13) [60]. To assess the likelihood of these complexes forming, we calculated the Gibbs free energy (ΔG). In thermodynamic terms, ΔG or the binding affinity of a complex provides a quantitative measure of interactions within a cell under specific conditions [95]. The results demonstrated that all TLR-MEV docked complexes were energetically favorable, as indicated by the negative ΔG values and multiple intermolecular contacts (see Table 10). This confirms that our MEV can effectively induce, bind, and sustain adequate immunological interactions.

The C-ImmSim server was employed to model the immune response elicited by our MEV. *In-silico* vaccine design often uses a three-dose regimen, with each dose spaced four weeks apart. This interval is based on immunological principles and empirical evidence from vaccine studies [96]. It allows the immune system sufficient time to process the antigen, develop a memory response, and maintain the initial immune activation. Research supports that this schedule enhances the immune response and provides long-term protection. The simulation results correspond well with observed real-world immunological responses [65, 96].

In our vaccine model, humoral responses were notably stronger after booster doses, with IgM levels surpassing those of IgG (Fig 14a). In the context of *in-silico* vaccine design, these levels are critical indicators of the immune response elicited by the vaccine. IgM antibodies, which are produced first during initial antigenic exposure, are associated with acute or early-stage infections and indicate that the immune system is encountering the antigen for the first time. Conversely, IgG antibodies are generated in a later phase, reflecting prior exposure or a secondary immune response [97, 98]. Although elevated IgM levels are effective for initiating an immune response, they may not fully demonstrate the potential for long-term immunity. In contrast, higher IgG levels are generally more indicative of sustained immune protection [97, 98]. The observed pattern of IgM and IgG responses in our vaccine model suggests its capacity to transition from a short-term immune response to the development of immune memory, which is essential for long-term protection.

B-cells are known to play a crucial role in antibody production and the generation of protective immunity [99]. Upon administration and repeated exposure to the MEV, there was a

significant elicitation of B-cell isotypes and B-cell memory cells, which endured for an extended period (see Fig 14a). Furthermore, the MEV administration triggered an adequate response from CD4+ T-cells and CD8+ T-cells. T-cell subsets are vital components of adaptive immunity in *Salmonella* infection, crucial for clearing primary infection and resisting subsequent challenges [99]. Given *Salmonella's* intracellular nature, recent research underscores the collaboration between CD4+ T-cells and B-cell responses in combating *Salmonella* infection [100], as demonstrated by our MEV simulations. There was a concurrent production of NK cells, dendritic cells, and macrophages following the initial dose of the MEV. Significant levels of cytokines, including IFN-γ, TGF-β, IL-2, IL-10, and IL-12, were elicited and maintained (Fig 16b). The overall activity of our MEV suggests that the selected epitopes effectively stimulated the necessary immunological responses for *Salmonella* protection and clearance post-infection.

In mammalian expression systems, codon optimization involves increasing the proportion of preferred mammalian codons in target genes. The notably high GC content (70.78%) of the enhanced MEV protein sequence and a CAI score exceeding 0.95 indicate the likely efficient expression of our target genes in humans. Codon usage in humans is generally higher than in most non-mammalian species, with more than 95% GC3 content. Remarkably, GC-rich genes exhibit several-fold to over 100-fold higher expression efficiency compared to GC-poor counterparts [101].

Selecting the right plasmid is integral to the success of both *in-vitro* and *in-vivo* studies in vaccine design. A well-chosen plasmid can enhance transfection efficiency, antigen expression, immune response, and safety, ultimately leading to a more effective and dependable vaccine. For this study, we chose the pcDNA3.1 plasmid due to its highly versatile backbone, which supports the expression of a wide range of proteins (Fig 18). This versatility makes it suitable for various vaccine candidates and experimental setups. Additionally, the design of pcDNA3.1 promotes stable expression in mammalian cells, which is essential for maintaining consistent and reliable antigen production [18, 50, 68–71]. Overall, the pcDNA3.1_CT-GFP plasmid offers high expression levels, ease of tracking, and adaptability, making it an invaluable tool in vaccine development.

## 5.0 Conclusion

Several conclusions can be drawn from this study. The selected MHC-I and MHC-II epitopes demonstrated antigenicity, were non-allergenic, and non-toxigenic, offering a 94.91% population coverage when used in MEV construction. Notably, the epitopes 'LAPDRRVEI' and 'IEYAITPEI' were identified as IFN-γ inducers, and all chosen epitopes for the MEV exhibited 100% conservation. The MEV protein construct comprised of 16.67% alpha-helices, 8.87% extended beta strands, 4.55% beta turns, and 69.91% random coils, and was found to be soluble, thermostable, and basic with a molecular weight of 47.343 kDa. Molecular dynamics simulations of the docked MEV and TLR complexes suggested that these complexes were energetically feasible and exhibited high binding affinity. The eigenvalues indicated a lower likelihood of deformation during immunological activity. The MEV also induced robust IgM and IgG responses, alongside their isotypes, and activated CD4+ and CD8+ T-cells, NK cells, dendritic cells, macrophages, and cytokines such as IFN-γ, TGF-β, IL-2, IL-10, and IL-12. While these *in-silico* results are promising, they warrant further validation through *in-vitro*, and *in-vivo* studies.

## 6.0 Limitations of the study

This study employed an *in-silico* methodology exclusively for designing the multiepitope peptide vaccine construct and validating the simulated immune responses. Despite the promise of

*in-silico* vaccine design, its effectiveness is limited by the current state of algorithms and computational methods, which are continually evolving. *In-silico* results require further validation through *in-vitro* (laboratory-based) and *in-vivo* (animal or human) experiments, as simulations based on computational models and population studies may not accurately reflect real-world scenarios. This is due to the complex nature of immune system interactions and individual variability. Additionally, regulatory bodies often demand extensive *in-vivo* evidence before accepting *in-silico* data as sufficient for vaccine approval. Computational vaccine design also raises ethical concerns regarding data privacy and the use of genetic information.

## Acknowledgments

The authors are thankful to the staff of the Bacterial Vaccine Production Department at the National Veterinary Research Institute, Vom, Nigeria, for their technical support during this research.

## Author Contributions

**Conceptualization:** Elayoni E. Igomu, Paul H. Mamman.

**Data curation:** Elayoni E. Igomu, David O. Ehizibolo.

**Investigation:** Elayoni E. Igomu, Paul H. Mamman.

**Methodology:** Elayoni E. Igomu, Paul H. Mamman, Jibril Adamu, Abubarkar O. Woziri, David O. Ehizibolo.

**Resources:** Maryam Muhammad.

**Software:** Elayoni E. Igomu, Abubarkar O. Woziri.

**Supervision:** Paul H. Mamman, Jibril Adamu, David O. Ehizibolo.

**Writing – original draft:** Elayoni E. Igomu.

**Writing – review & editing:** Elayoni E. Igomu, Paul H. Mamman, Jibril Adamu, Maryam Muhammad, Abubarkar O. Woziri, Manasa Y. Sugun, John A. Benshak, Kingsley C. Anyika, Rhoda Sam-Gyang, David O. Ehizibolo.

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
