## [Decision Letter · Decision Letter 0]

30 Jul 2024

PONE-D-24-23783Immunoinformatics design of a novel multiepitope vaccine candidate against non-typhoidal salmonellosis caused by Salmonella Kentucky using outer membrane proteins A, C, and FPLOS ONE

Dear Dr. Igomu,

Thank you for submitting your manuscript to PLOS ONE. After careful consideration, we feel that it has merit but does not fully meet PLOS ONE’s publication criteria as it currently stands. Therefore, we invite you to submit a revised version of the manuscript that addresses the points raised during the review process.

We have now received the reports from our reviewers, and after careful consideration, we have decided that your manuscript requires major revisions before it can be considered for publication. The reviewers have provided detailed feedback that we believe will significantly enhance the quality and clarity of your manuscript.

Key areas requiring revision include:

1. Sequence Diversity: Provide detailed analysis and presentation of the sequence diversity of the outer membrane proteins (OmpA, OmpC, and OmpF).

2. Language and Structure: Address significant grammatical errors and improve the manuscript's clarity and organization. Consider using professional editing services.

3. Methodology: The methods section should include additional technical details and present a figure explaining the stepwise flow of the methodology.

4. Experimental Data: Discuss potential challenges and limitations of the immunoinformatic approach and provide a rationale for the chosen methodologies. Cite references that have used similar approaches with in vivo evaluations.

Addressing these comments will significantly strengthen your manuscript. Please submit a revised version of your manuscript with a detailed response to each reviewer's comment. If you have any questions or need further clarification, please do not hesitate to contact us.

We look forward to receiving your revised manuscript.

Kind regards,

Prashant Sharma, Ph.D.

Academic Editor

PLOS ONE

Reviewers' comments:

Reviewer's Responses to Questions

**Comments to the Author**

1. Is the manuscript technically sound, and do the data support the conclusions?

Reviewer #1: Partly

Reviewer #2: Partly

Reviewer #3: Partly

2. Has the statistical analysis been performed appropriately and rigorously? 

Reviewer #1: Yes

Reviewer #2: N/A

Reviewer #3: N/A

3. Have the authors made all data underlying the findings in their manuscript fully available?

Reviewer #1: Yes

Reviewer #2: Yes

Reviewer #3: Yes

4. Is the manuscript presented in an intelligible fashion and written in standard English?

Reviewer #1: Yes

Reviewer #2: No

Reviewer #3: Yes

5. Review Comments to the Author

Reviewer #1: The authors employed a bioinformatics approach to analyze the physicochemical properties of three outer membrane proteins of Salmonella Kentucky and subsequently predicted their antigenicity and immunogenicity. This method demonstrates an effective and predictive strategy for developing a pipeline for various vaccines. Furthermore, by considering the HLA types of different ethnic groups, the study assesses whether these targets can be used universally for vaccines or diagnostics. However, a significant concern is the lack of clarity regarding the sequence diversity of the outer membrane proteins (OmpA, OmpC, and OmpF) analyzed throughout the paper, which limits the efficacy of the proposed target regions.

1. The authors collected outer membrane sequences with 99% to 100% identity using BLAST, resulting in a narrow range of antigenic sequences that may not be suitable for protection against various Salmonella Kentucky strains. The authors should describe the number of "complete coding sequences (CDS)" as a percentage.

2. Additionally, the candidate regions cover variation of OmpA, OmpC, and OmpF in all Salmonella Kentucky strains? The diversity of OmpA, OmpC, and OmpF should be analyzed to understand the sequence diversity and efficacy of the vaccine/diagnostic candidates. To address this, a maximum likelihood phylogenetic tree based on codon alignments using a non-redundant CDS collection of the Omp genes should be presented. Authors should confirm whether clusters in the phylogenetic tree correlate with amino acid sequence variation in the candidate regions.

Reviewer #2: The manuscript presents a substantial amount of valuable data that is highly relevant and can contribute significantly to the field of vaccinology. However, in my opinion the paper has some

Shortcomings and the manuscript's current structure and clarity need significant improvement.

Major grammatical errors present, the manuscript needs language editing. Consider using professional editing services. Method section needs to be more systematically written with sufficient technical details to be more detailed and clearly described. While the data is extensive, the results section is challenging to follow.

Specific comments

1. The manuscript lacks proper punctuation, including brackets and full stops eg; pg 4, line 86, pg5, line 111

2. Pg4, line 90: what are the various risk factors?

3. Italicise all species names

4. Please ensure that capitalization is used correctly eg: pg6, line 128: Human

5. Good to have a figure explaining the stepwise flow of the methodology.

6. Use the right terminology, pg 6, line 113: BLOSUM62 is substitution matrix.

7. Was the signal peptide identified?

8. Pg 5, line 97: FASTA format sequences in aa or nucleotide?

9. Describe in detail the alleles used for T‑Cell epitope prediction analysis.

10. What is the IC50 threshold values set?

11. Pg 6, line 116: Epitope Analysis Resource from Immune Epitope Database (IEDB) server.

12. Why ab initio method was preferred for tertiary structure prediction?

13. Table 2 can be moved to supplementary file.

14. The number of OMP sequences selected varies, OMPF has only 15 sequences?

15. Table 3: can differentiate the epitopes by the protein (OMP A, C and F)

16. Table 6; positive (typo), differentiate the OMPs.

17. Resolution of the figures could be improved.

18. Figure legends/captions should include a clear and concise description of the figure. eg Figure 7: Docking of the 3D tertiary structural models (list the colours of the MEV and TLRs)

Reviewer #3: PONE-D-24-23783

General comment:

The research article "Immunoinformatics design of a novel multiepitope vaccine candidate against non-typhoidal salmonellosis caused by Salmonella Kentucky using outer membrane proteins A, C, and F". The research demonstrates a well-structured approach to vaccine design using bioinformatics and immunoinformatics techniques. The study created a novel MEV construct from OMP A, C, and F with the inclusion of fliC protein as an adjuvant from Salmonella typhimurium and characterized it using various software. The MEV was identified as having 14 CD8+ and 7 CD4+ T-cell epitopes, along with immunogenic B-cell epitopes. The vaccine construct demonstrated high structural quality, favorable physicochemical properties, and stability in molecular docking and dynamic simulations. Immune simulations indicated significant dose-dependent immune responses of IgG and IgM. The MEV is predicted to be non-toxigenic and non-reactogenic in nature.

Specific comments:

The authors used immunoinformatics tools to construct a potential MEV candidate for Salmonella Kentucky and characterized it with various software / in-silico tools to predict interferon-gamma (IFN-γ) 420 induction capabilities, discontinuous B-cell epitopes, secondary and tertiary structure, Protein-protein docking and protein-protein binding affinity, Molecular dynamics, and Immune Simulation. The authors also cited various reference (38, 47, 48, 54, 62, 63, 66, 73, 77, 78, 81, 82) which has used similar approach of immunoinformatics to design vaccine for various infectious diseases. However, the finding does not provide experimental data to support the theoretical findings, which is crucial for translating the research into a viable vaccine candidate. This study could benefit from a more detailed discussion of the potential challenges and limitations of this approach. Additionally, it is important to cite the reference who used the immunoinformatics approach to design a vaccine candidate and evaluated it in small animals to support in-silico Vs in-vivo results.

Section 2.8:

• Line no 320 - “Ensuring the efficacy of the vaccine model during cloning and expression is crucial for its in-vitro production” should be changed to “Ensuring the efficacy of the vaccine model during cloning and expression is crucial for its in-vivo production”

• It is not clear why authors in this study performed codon optimization for a vaccine construct for a mammalian host. The MEV proposed is a bacterial-derived chimeric protein construct that can be easily expressed in a simple host like E. coli. This poses a question on the rationale for the construction of MEV codon.

Section 2.19:

• It is not clear why the authors introduced 6xHis tag on the C-terminal of the MEV in this study as all the experiments were in silico. It is recommended to avoid His tags in vaccine candidate when predicting their immune response using the database. The MEV containing His tag may potentially lead to the generation of anti-histamine antibodies.

• Authors have also not described whether any antibiotic marker is used in a plasmid construct.

Section 3.6 and discussion section:

The authors have not described the rationale for the order of linkage B cell, HTL and CTL epitope peptide linkage in a MEV construct. Was it random or a particular order of different peptide sequences of epitopes were followed to construct MEV?

Section 3.12 and discussion section:

• The immune simulation predicted the humoral responses 565 for both IgM and IgG were more pronounced after booster doses compared to the initial dose, with IgM levels being higher than IgG (and its isotypes). A detailed discussion would be beneficial to describe the significance of a higher level of IgM as compared to IgG and what it means.

• A detailed discussion on the rationale of three doses and dose intervals while performing an immune stimulation.

Section 5.0:

Line no. 732: The author claimed MEV to be thermostable. However, the rationale for this claim is neither mentioned in the result section nor described in the discussion section. What parameters or scores were responsible for predicting good solubility and thermos-stability?

6. PLOS authors have the option to publish the peer review history of their article (what does this mean?). If published, this will include your full peer review and any attached files.

Reviewer #1: No

Reviewer #2: No

Reviewer #3: **Yes: **Ruchirkumar Pansuriya

---

## [Author Response · Author response to Decision Letter 0]

6 Sep 2024

PONE-D-24-23783

Key areas addressed as requested by the academic editor

1. To better understand the antigenic diversity for the different classes of OMPs (OmpA, OmpC, and OmpF), the percentage conservation of the amino acid sequence has been computed using the Phylogeny.fr server (Section 2.2 line 121), the results are in section 3.1 and table 2. The phylogenetic relationships, based on the maximum likelihood method, have been computed as recommended (Fig 2).

2. The language and structure have been greatly improved as recommended. Several paragraphs have been restructured to enhance clarity, professional tone, and flow. Attention has been given to punctuation details, abbreviations, table headers, and legends.

3. A flow chart explaining the methods in a stepwise manner has been included in the manuscript as recommended (Fig 1). The methodologies were improved upon, and the rationale behind the methods used was stated. You can find them in sections 2.2 (Lines 121 to 129 ), 2.9 (lines 223 to 226), section 2.17 (Lines 333 to 344), section 2.18 (lines 349 to 354 ), and 2.19 (lines 376 to 383)

4. The potential challenges and limitations of the use of the immunoinformatics approach have been provided (Section 6.0). Finally, I have incorporated references to studies that have used an in-silico approach to vaccine design with in-vivo evaluations (References 18, 35, 47, 69, 70, and 71)

Response to reviewers 

Response to Reviewer # 1

1. Outer membrane proteins (OMPs) are highly conserved and identical within each class of protein in Salmonella, regardless of the serovar (as stated in lines 80 to 83). However, the percentage conservation across different classes of amino acid sequences has been computed and presented in section 3.1 (lines 394 to 403) and Table 2. 

To avoid ambiguity, section 2.1 has been rephrased to clarify intent. The reason for blasting at 99 % to 100% for each class of OMP in our study was majorly to narrow down our selection criteria, to ensure all selected proteins were those isolated from Salmonella Kentucky. This does not in any way narrow the antigenic range within each class of protein.

2. To better understand the antigenic diversity for the different classes of OMPs (OmpA, OmpC, and OmpF), a phylogenetic tree based on maximum likelihood has been computed and added as recommended (Fig 2). 

Response to Reviewer # 2

1. Punctuations have been improved, typographical errors corrected and the right terminologies noted.

2. A statement on the various risk factors that make Africa prone to the emergence of AMR clones has been added (Lines 90 to 93).

3. All species names have been italicized 

4. The IC50 threshold value chosen has been stated (Line 160).

5. Tables have been updated to differentiate epitopes by protein classes (Tables 3 and 6).

6. The use of “amino acid” in FASTA format has been clearly stated (Section 2.1)

7. Legends for figures 3, 4, 5, 6, and 12–16, have been updated to improve clarity and conciseness.

8. Figure 9, formerly Figure 7, depicting the dock complex between TLRs and MEV, has been updated to include the color coding between MEV and TLRs, as recommended.

Response to Reviewer # 3

1. References to studies that used the immunoinformatics approach to design a vaccine candidate and evaluated it in small animals to support in-silico Vs in-vivo results have been included (18, 35, 47, 69, 70, 71).

2. The rationale for the choice of plasmid and the choice of dosing regimen has been explained in the manuscript as advised and references that have used similar approaches are cited (Section 2.17, Line 333–344; Section 2.18, line 349–355). References 39, 65, 66, 67, 68.

3. Codon optimization was performed for mammalian hosts because the vaccine was designed for humans. In vitro experiments (e.g., transfection studies) are often conducted in mammalian cell lines for human vaccines. Additionally, most in vivo experiments are performed in laboratory animals like mice, rats, and monkeys, which are mammals. Studies that employed mammalian codon optimization and or, the pcDNA3.1 plasmid, have been referenced in the manuscript (sections 2.18 and 2.19), specifically in references 18, 35, 68, 69, and 70.

3. The immune simulation study was done using the amino acid (AA) sequence of the multi-epitope vaccine construct without the 6xHis tag AA sequence. The inclusion of the 6xHis tag was during cloning into the plasmid after the immune simulation had been conducted. Studies with similar approaches have been referenced (Reference 47). The addition of the 6xhis tag to the protein before cloning was to make it easy to track the protein during expression and purification processes when conducting in-vitro experiments. 

The detection of anti-His antibodies is what makes it easier to track the proteins of interest during expression and purification processes. Finally, 6xHis tag usually does not interfere with the protein’s native structure or function.

4. Most in-silico designs employ plasmid with antibiotic markers. They are utilized to identify and successfully select transformants, particularly in cloning, gene expression, and genetic engineering. This study and other studies referenced in this manuscript employed plasmids that carry antibiotic markers. The AmpR (ampicillin-resistant) gene carried on our plasmid was the marker (Fig 18).

5. The rationale for the order of linkage of the epitopes for the construction of the MEV has been included in the manuscript (Section 2.9, lines 223–227).

6. A detailed discussion on the implication of the high IgM has been included in the manuscript (Lines 762 – 773).

7. The physicochemical properties of the vaccine that make it thermostable and soluble are available in the results (Section 3.7, Lines 488 to 490 for solubility; and lines 491 to 494 for aliphatic index). These parameters and scores have been quoted in the discussion (lines 735– 739).

---

## [Decision Letter · Decision Letter 1]

9 Oct 2024

PONE-D-24-23783R1Immunoinformatics design of a novel multiepitope vaccine candidate against non-typhoidal salmonellosis caused by Salmonella Kentucky using outer membrane proteins A, C, and FPLOS ONE

Dear Dr. Igomu,

Thank you for submitting your manuscript to PLOS ONE. After careful consideration, we feel that it has merit but does not fully meet PLOS ONE’s publication criteria as it currently stands. Therefore, we invite you to submit a revised version of the manuscript that addresses the points raised during the review process.

Both reviewers have recommended minor revisions. Given this, I would kindly ask you to carefully review the manuscript for any areas that may benefit from further clarification or minor improvements, including correcting typographical errors, enhancing the explanation of key concepts, or addressing any formatting inconsistencies. Authors should ensure the text is as clear and polished as possible before resubmission.

We look forward to receiving your revised manuscript.

Kind regards,

Prashant Sharma, Ph.D.

Academic Editor

PLOS ONE

Journal Requirements:

Reviewers' comments:

Reviewer's Responses to Questions

**Comments to the Author**

1. If the authors have adequately addressed your comments raised in a previous round of review and you feel that this manuscript is now acceptable for publication, you may indicate that here to bypass the “Comments to the Author” section, enter your conflict of interest statement in the “Confidential to Editor” section, and submit your "Accept" recommendation.

Reviewer #1: All comments have been addressed

Reviewer #3: All comments have been addressed

2. Is the manuscript technically sound, and do the data support the conclusions?

Reviewer #1: Partly

Reviewer #3: Yes

3. Has the statistical analysis been performed appropriately and rigorously? 

Reviewer #1: I Don't Know

Reviewer #3: Yes

4. Have the authors made all data underlying the findings in their manuscript fully available?

Reviewer #1: Yes

Reviewer #3: Yes

5. Is the manuscript presented in an intelligible fashion and written in standard English?

Reviewer #1: Yes

Reviewer #3: (No Response)

6. Review Comments to the Author

Reviewer #1: I understand the authors' point. My question is whether the vaccine candidate developed by the authors could also be effective against other strains of Salmonella Kentucky, which sequences are available in NCBI but were not discussed in this article.

Reviewer #3: (No Response)

7. PLOS authors have the option to publish the peer review history of their article (what does this mean?). If published, this will include your full peer review and any attached files.

Reviewer #1: No

Reviewer #3: No

---

## [Author Response · Author response to Decision Letter 1]

4 Nov 2024

PONE-D-24-23783R1

Key areas addressed as requested by Journal

I have carefully reviewed all references cited in this manuscript to confirm their accuracy and completeness and can confirm that no retracted articles have been cited or identified.

Response to reviewers 

Response to Reviewer # 1

Yes. The developed vaccine candidate is expected to be effective against all strains of Salmonella Kentucky, including those with and without available sequences in the NCBI database, as the outer membrane protein sequences used in this study incorporate highly conserved regions.

Response to Reviewer # 3

Response to Specific Comment (2.18 Codon Adaptation):

The authors appreciate the insightful feedback regarding glycosylation concerns for the MEV construct expressed in mammalian cells. 

The authors recognize the potential for glycosylation at certain amino acid residues. Although the in-silico modeling did not integrate genetic interventions to address glycosylation, this issue can be effectively addressed during in-vitro experimentation:

1. Post-Expression Enzymatic Treatment: In-vitro assays can utilize PNGase F (N-Glycosidase F) to remove any N-linked glycans after expression enzymatically. While this does not inhibit glycosylation during protein synthesis, it ensures that the final product remains glycosylation-free, making it a closer structural analog to the bacterial protein.

2. Use of Glycosylation-Deficient Cell Lines: Certain cell lines have been genetically modified to lack glycosylation pathways, such as CHO-K1 cell lines, which we intend to select for this purpose. Using these glycosylation-deficient cell lines minimizes or eliminates unintended glycosylation during protein expression.

Given that this manuscript focuses exclusively on in-silico findings, the authors have not detailed the use of CHO-K1 cell lines in this research phase. We anticipate that these in-vitro techniques will adequately address glycosylation concerns in future experimentation phases.

---

## [Editor Report · Decision Letter 2]

7 Nov 2024

Immunoinformatics design of a novel multiepitope vaccine candidate against non-typhoidal salmonellosis caused by Salmonella Kentucky using outer membrane proteins A, C, and F

PONE-D-24-23783R2

Dear Dr. Igomu,

We’re pleased to inform you that your manuscript has been judged scientifically suitable for publication and will be formally accepted for publication once it meets all outstanding technical requirements.

Kind regards,

Prashant Sharma, Ph.D.

Academic Editor

PLOS ONE
---

## [Editor Report · Acceptance letter]

11 Nov 2024

PONE-D-24-23783R2 

PLOS ONE

Dear Dr. Igomu, 

I'm pleased to inform you that your manuscript has been deemed suitable for publication in PLOS ONE. Congratulations! Your manuscript is now being handed over to our production team.

Kind regards, 

on behalf of

Dr. Prashant Sharma 

Academic Editor

PLOS ONE